# PAL-AI reveals genetic determinants that control poly(A)-tail length during oocyte maturation, with relevance to human fertility

Kehui Xiang [1,2,3] ✉ & David P. Bartel [1,2,3] ✉

In oocytes of mammals and other animals, gene regulation is mediated primarily through changes in poly(A)-tail length. Here, we introduce PAL-AI, an integrated neural network machine-learning model that accurately predicts tail-length changes in maturing oocytes of frogs and mammals. We show that PAL-AI learned known and previously unknown sequence elements and their contextual features that control poly(A)-tail length, enabling it to predict tail-length changes resulting from 3′-untranslated region single-nucleotide substitutions. It also predicted tail-length-mediated translational changes, allowing us to nominate genes important for oocyte maturation. When comparing predicted tail-length changes in human oocytes with genomic datasets of the All of Us Research Program and gnomAD, we found that genetic variants predicted to disrupt tail lengthening have been under negative selection in the human population, thereby linking mRNA tail lengthening to human female fertility.

Fully grown vertebrate oocytes, which are arrested at prophase I of meiosis, have minimal transcriptional activity due to the condensation of chromosomes. With little potential for transcriptional control, post-transcriptional control, particularly translational control, plays a major role in regulating gene expression during the subsequent processes of oocyte maturation and early embryonic development, after which the zygotic genome is transcriptionally activated[1–3]. Proper progression through this period of oocyte maturation and early development requires translational upregulation of select maternally deposited mRNAs, such as *MOS* and *CCNB1*[4,5], which encode essential proteins for meiosis, and *TPRX1/2/L*[6], which encode transcription factors necessary for zygotic genome activation. Leveraging the strong coupling between poly(A)-tail length and translational efficiency during these developmental stages[7–14], translation activation of these mRNAs is achieved by extension of their poly(A) tails[4,15,16], which enables them to compete better for the binding of limited poly(A)-binding proteins and subsequent recruitment of translation initiation factors[11]. Thus, disruption of either translational or poly(A) tail-length changes in these maturing oocytes or early embryos can cause developmental arrest in frogs, mice, and humans[6,17,18]. Genetic alterations to these processes have also been linked to human female infertility[19–21].

Control of poly(A)-tail length during oocyte maturation of vertebrate animals is a tug-of-war between the default transcriptome-wide deadenylation and the mRNA-specific cytoplasmic polyadenylation[10,13,14,22–24]. The specificity of cytoplasmic polyadenylation relies on two sequence elements within mRNA 3′ untranslated regions (UTRs): a cytoplasmic polyadenylation element (CPE, motif UUUUA), which is recognized by the CPEB1 protein, and a polyadenylation signal (PAS, motif AWUAAA, where W is either A or U), which is recognized by the cleavage and polyadenylation specificity factor (CPSF) complex[14,25–28]. Contextual features of these two sequence elements, including their relative positions, flanking nucleotides, and structural accessibility, as well as the number of each element within the 3′ UTR, influence the extent of cytoplasmic polyadenylation[14]. Although several other sequence elements have been proposed to influence poly(A)-tail length, their effects are usually small, indirect, and do not appear to impact more than a handful of mRNAs[14,27,29,30].

Despite advances in our knowledge of cis-acting elements that control cytoplasmic polyadenylation, a quantitative understanding of poly(A) tail-length changes for different mRNAs during oocyte maturation is still lacking. A sequence-based predictive model would provide further insights into known functional sequence elements and

[1]Howard Hughes Medical Institute, Cambridge, MA, USA. [2]Whitehead Institute for Biomedical Research, Cambridge, MA, USA. [3]Department of Biology, Massachusetts Institute of Technology, Cambridge, MA, USA. ✉e-mail: kxiang@wi.mit.edu; dbartel@wi.mit.edu

enable the discovery and evaluation of additional motifs or regulatory principles. For example, such a model would enable large-scale assessment of the consequences of genetic mutations on tail-length control as well as translational regulation, thus providing an opportunity to nominate disease-contributing variants that are otherwise difficult to identify in genome-wide association studies. Therefore, we set out to develop a machine-learning model that accurately predicts poly(A) tail-length changes in humans and other vertebrate animals.

## Results

### An integrated neural network model predicts tail-length changes during frog oocyte maturation

To predict tail-length changes during frog oocyte maturation directly from mRNA sequences, we initially developed a multiple linear regression model based on $k$-mer (sequence motif length of $k$) compositions within 3′ UTRs. Given the importance of both the number and position of CPE and PAS elements in influencing tail-length changes[14], we defined two features for each $k$-mer of varying lengths: 1) a count metric, specifying the total number of a $k$-mer within the 3′ UTR; 2) a positional metric, calculated as the sum of the inverse distances between each $k$-mer and the 3′ end. We performed training and testing in a 10-fold cross-validation procedure on frog endogenous mRNA tail-length changes, comparing between 0 h and 7 h post-progesterone treatment[14] (Fig. 1a). Test-set predictions from each fold were concatenated and compared to the measured values (Fig. 1b). We tested different $k$-mer and 3′ UTR lengths as inputs and conducted extensive hyperparameter tuning (Supplementary Fig. 1a). The best model performed moderately well and explained approximately 40% of the variance of measured tail-length changes (Fig. 1c; Spearman correlation coefficient $R_s = 0.64$; Pearson correlation coefficient $R_p = 0.63$). Notably, the CPE and the PAS, two known motifs mediating cytoplasmic polyadenylation, were among the top-ranked features in both the count and the positional metrics (Supplementary Fig. 1b, c).

Although the linear model identified the essential motifs for poly(A)-tail lengthening, it under-predicted tail-length changes for most mRNAs whose tails were extended by >30 nt. Inspired by recent advances in deep learning, we developed a neural network model, termed poly(A)-tail length AI (PAL-AI) (Fig. 1d, Supplementary Fig. 1d, e), which integrates both convolutional and recurrent neural network components, using an architecture similar to but simpler than that of Saluki[31]. PAL-AI was trained and tested using the sequences of endogenous mRNAs, and the same 10-fold cross-validation strategy as used for the linear model. Combinations of different hyperparameters were tested extensively for the model to achieve the best performance (Supplementary Fig. 2a, b) without overfitting (Supplementary Fig. 2c, d). The final trained model predicted tail-length changes with high accuracy, significantly out-performing the linear model and explaining more than 67% of the variance of measured values (Fig. 1e; $R_s = 0.82$, $R_p = 0.82$).

To examine the impact of input sequence length on prediction accuracy, we trained PAL-AI models on various regions of the 3′ UTR and compared Pearson correlation coefficients between predicted and measured values. A model trained exclusively on the last 100 nt of 3′ UTRs already explained over 40% of the variance (Fig. 1f; average $R_p = 0.64$). Model performance improved with increasing 3′ UTR length up to 2000 nt, beyond which no significant gains were observed (Fig. 1f, g, Supplementary Data 5). In contrast, when the last 100 nt were excluded from the last 1000 nt of 3′ UTRs for the input, the predictive power declined markedly (Fig. 1f; average $R_p = 0.40$), supporting the idea that the 3′-most regions of 3′ UTRs impart the strongest effect on tail-length changes[14].

We further explored whether incorporating additional features, such as sequences from coding regions and base-pairing probabilities predicted by RNAfold[32], could improve the overall model performance. However, models including these features showed no significant improvement over those trained on equally-sized 3′ UTR-only inputs (Fig. 1f, g), implying that these features, at least as parameterized in our model, contribute minimally to tail-length changes. Nevertheless, among all configurations, the model trained on the last 2000 nt of the 3′ UTR plus the coding region performed the best with the shortest input length (Fig. 1f, g, Supplementary Data 5) and was therefore used for subsequent analyses unless otherwise specified. We also tested a variant model architecture, in which the convolutional module was replaced with a ResNet block, a design that has demonstrated superior performance in many deep-learning applications[33,34]. This alternative, although at times achieving comparable accuracy, was less consistent (Supplementary Fig. 2e) and required substantially more parameters and computational resources. For this reason, we did not further optimize this ResNet-based model.

### PAL-AI learned motifs and contextual features that regulate poly(A)-tail length

To elucidate the principles learned in the training of PAL-AI, we performed in silico mutagenesis and examined the impact of single-nucleotide mutations on predicted poly(A) tail-length changes. Among all 6-mers, the loss of the PAS (AAUAAA) and CPE (UUUUA)-containing 6-mers led to the largest decrease in tail-length changes (Fig. 2a). Sequence logos generated from the most impacted 8-mers also matched the CPE and the PAS (Fig. 2b). Similarly, when examining the consequence of gaining of 6-mer motifs, those motifs containing the PAS and CPE resulted in the greatest predicted tail-length increase (Supplementary Fig. 3a, b). To prevent the signal from spilling from a stronger motif to a weaker one, we reexamined tail-length changes associated with the loss of 6-mers. We identified the 6-mer associated with the largest mean tail-length change and then re-calculated the average tail-length changes for all remaining 6-mers after excluding all point substitutions that disrupted this top 6-mer. This process was repeated 14 times, each time excluding point substitutions that disrupted an additional 6-mer, thereby generating a ranked list of 15 different 6-mers most associated with predicted tail lengthening (Fig. 2c). Likewise, we used an analogous procedure to reexamine tail-length changes associated with the gain of 6-mers (Supplementary Fig. 3c). As expected, the top 6-mers identified in these analyses contained a PAS or CPE.

In addition to the PAS and CPE, our analyses also showed that PAL-AI recognized a different class of 6-mers, which were associated with modest but statistically significant tail lengthening (Fig. 2c, Supplementary Fig. 3c). Interestingly, these motifs each contained a UGU or GUU trimer, which might be recognized by other RNA-binding proteins such as CELF or DAZL in frog oocytes[35–37]. Another possibility is that some of these motifs, such as UGUUUU, which resemble the CPE, might be weakly bound by CPEB1. A third possibility is that the presence of these motifs correlates with tail lengthening but is not causal. To explore this latter possibility, we examined the efficacy of these motifs in arbitrary sequences used in a massively parallel reporter 3′ UTR library[14]. Because these arbitrary sequences were not biological, they were not susceptible to noncausal correlative associations, and thus any association of a motif with a tail-length change was presumably causal. Analysis of these reporters revealed an association between these UGU/GUU-containing 6-mers and tail lengthening, but it was small –– only a 0.64 nt increase on average for the top UGU/GUU-containing 6-mer UGUUUU in the absence of a CPE, which was substantially less than that of a CPE (17.7 nt on average)[14]. In the same library, a somewhat larger increase in tail length was observed for UGU and GUU (1.8 and 1.4 nt above an average effect of a 3-mer, respectively) when a CPE was also present in the 3′ UTR (Supplementary Fig. 3d). This added tail-lengthening effect was observed in all CPE-neighboring positions more than 2 nt from the CPE (Fig. 2d), suggesting that these motifs might play a small auxiliary role in tail lengthening beyond merely modulating the context of the CPE.

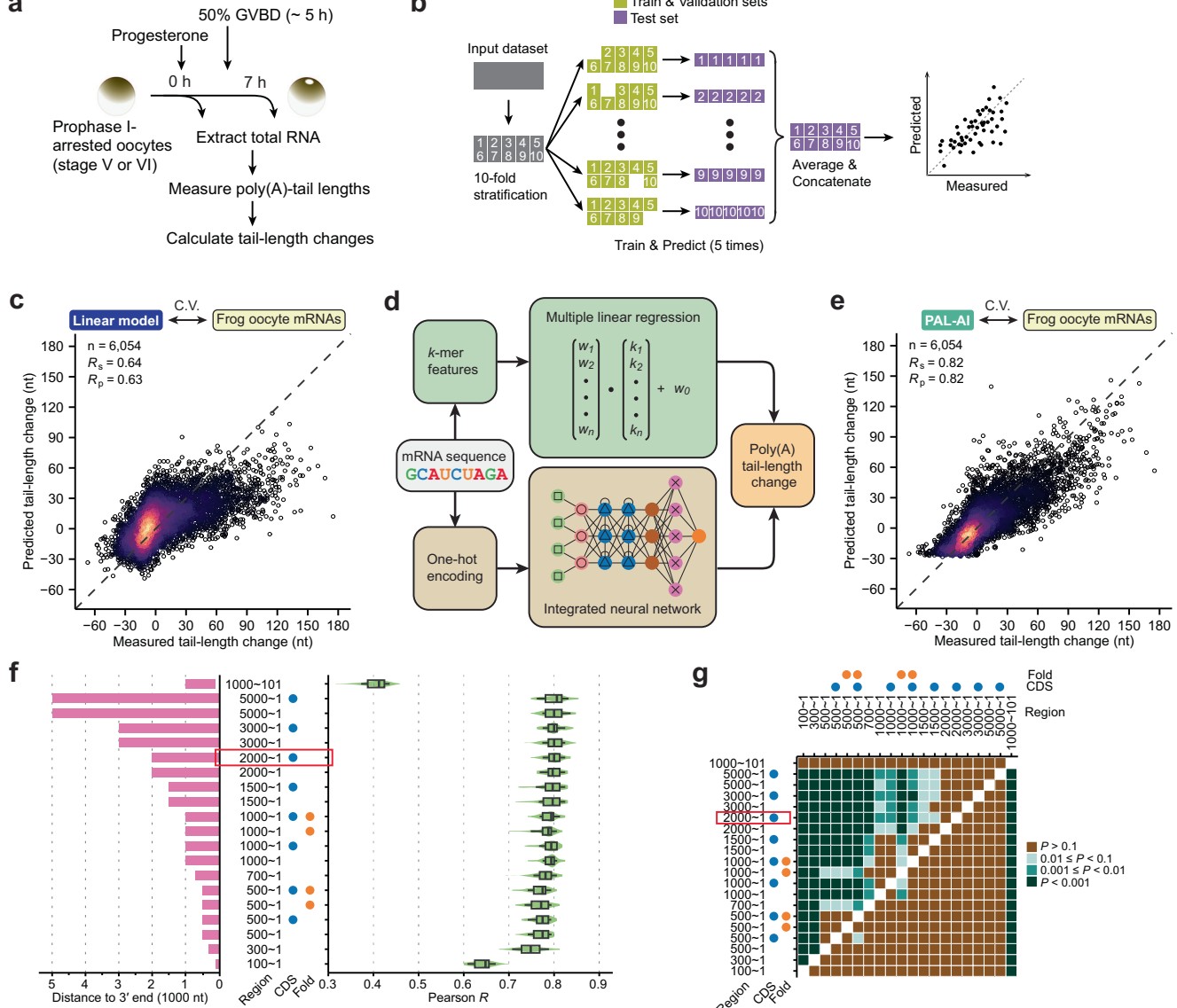

**Fig. 1 | Machine-learning models predict tail-length changes during frog oocyte maturation. a** Experimental scheme for examining poly(A) tail-length changes of frog mRNAs during oocyte maturation. Total RNA was extracted from oocytes before and after progesterone-induced germinal vesicle breakdown (GVBD), and changes in poly(A)-tail lengths were measured. **b** Schematic of the 10-fold cross-validation strategy used to train and test different machine-learning models in this study. Data were partitioned into training/validation and test sets, repeated across 10 different stratified folds. **c** Performance of the multiple linear regression model. Plotted are the tail-length changes predicted by the model as a function of the changes measured in frog oocytes between 0 h and 7 h post-progesterone treatment. Each point represents a unique poly(A) site of an endogenous mRNA. Colors indicate the density of points. C.V., cross validation. **d** Diagram outlining the two machine-learning models developed to predict poly(A) tail-length changes from mRNA sequences: a multiple linear regression

model and an integrated neural network (PAL-AI). **e** Performance of PAL-AI; otherwise as in (**c**). **f** Prediction performance of PAL-AI trained on different input regions of mRNAs or additional annotation features. Left: input sequence regions (bars) and additional features, i.e, coding sequence (CDS) or predicted pairing (fold), blue and orange dots, respectively. Right: distributions of $R_p$ values observed when comparing predicted and measured tail-length changes for test data held out during training. Ten-fold cross-validation of the model was repeated five times, generating 50 $R_p$ values. The red rectangle indicates the configuration chosen as the final model. Box and whiskers indicate the 10th, 25th, 50th, 75th, and 90th percentiles. **g** Pairwise comparison of input strategies used for PAL-AI based on prediction performance. Shown are binned $P$ values from one-sided t-tests, testing the alternative hypothesis that the mean $R_p$ value of the group indicated on the y axis is greater than that indicated on the x axis. The red rectangle indicates the configuration chosen as the final model.

Moreover, for some 3′ UTRs, the co-presence of multiple copies of these UGU/GUU-containing motifs might still contribute to detectable tail lengthening[14], perhaps through binding to CPEB1-interacting proteins, such as DAZL[38,39], which in turn could facilitate recruitment of CPEB1 to adjacent CPE-like motifs.

We carried out additional in silico mutagenesis by inserting either the CPE or the PAS at each position along the 3′ UTRs. As expected[14], PAL-AI predicted the strongest impact on polyadenylation when these elements were introduced near the 3′ ends of the 3′ UTRs (Fig. 2e). An

added CPE was most favorable when located near the PAS on either its 5′ or 3′ side (Fig. 2f), but not in the position most likely to disrupt the PAS, which is typically -17 nt from the end of the 3′ UTR (Supplementary Fig. 3e)[40], consistent with the positional effects determined from analyses of millions of reporters[14].

Although insertion of the canonical PAS variant AUUAAA moderately promoted polyadenylation (Fig. 2e), noncanonical PAS variants[41] had negligible effects on cytoplasmic polyadenylation (Supplementary Fig. 3f). In fact, mutating the canonical PAS AAUAAA

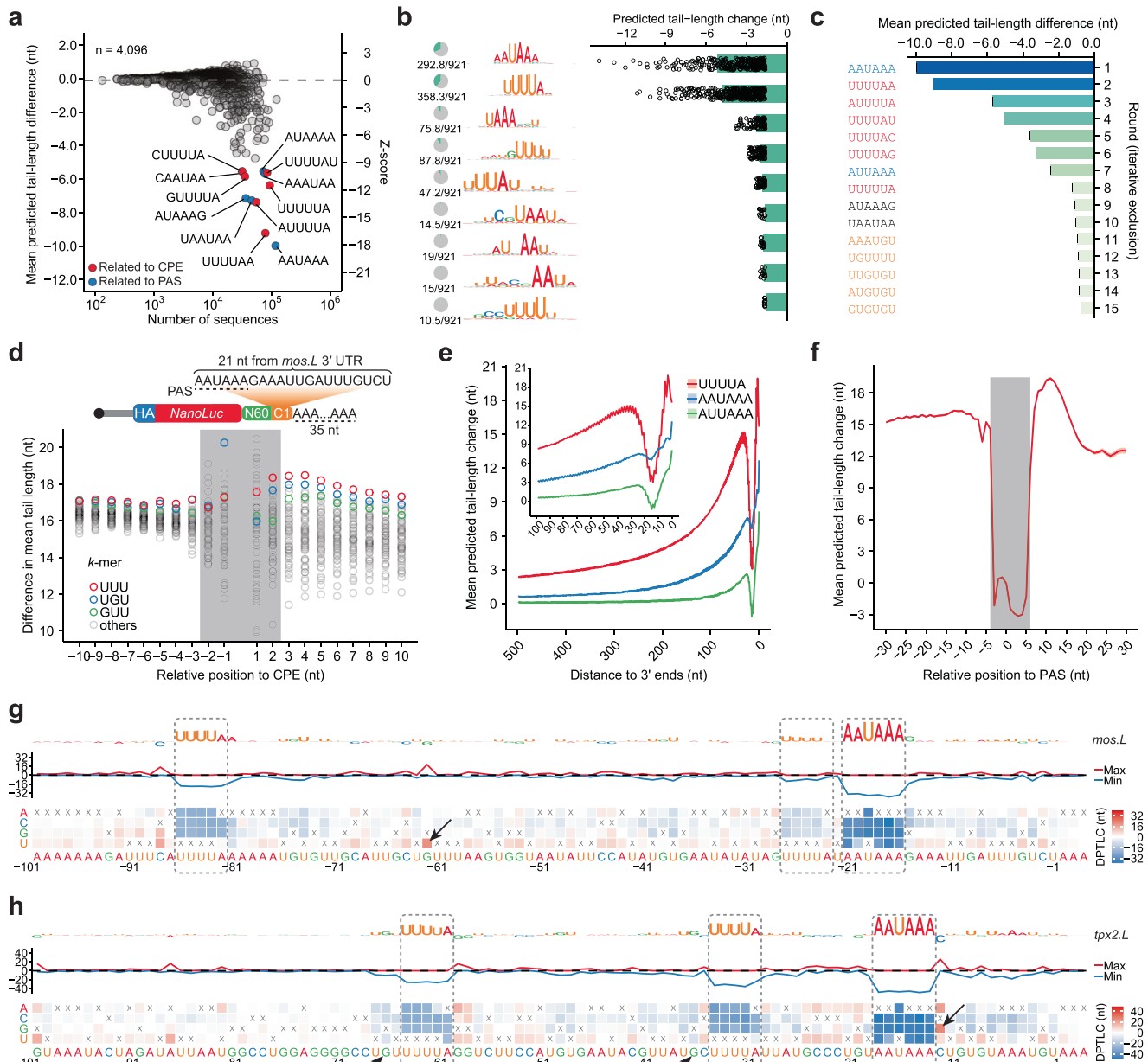

**Fig. 2 | PAL-AI learned sequence elements and contexture features important for tail-length control during oocyte maturation. a** PAL-AI-predicted consequences of motif loss. For each 6-mer, the mean difference in predicted tail-length change, comparing mutants and wild-type, is plotted as a function of the number of analyzed mutants. CPE- or PAS-related motifs are indicated (red and blue, respectively). **b** Sequence motifs associated with PAL-AI-predicted changes in tail length. Sequence logos were generated from 8-mers most associated with the largest differences in PAL-AI-predicted tail-length change upon 8-mer loss. Pie charts indicate fractions of 8-mers aligned to logos. Bar plots show mean differences; points represent individual 8-mers. **c** Top 6-mers associated with decreased predicted tail-length change from in silico mutagenesis, selected iteratively, with exclusion. Motif colors: CPE (red), PAS (blue), GUU/UGU (orange), others (black). Error bars, standard error. **d** Positional effects of 3-mers flanking a CPE. Plotted for each 3-mer are differences in mean tail length for mRNAs with that 3-mer at indicated positions relative to a CPE in the N60-PAS[mos] library[14]. The gray box indicates positions where 3-mers impact the CPE context[14]. **e** PAL-AI-predicted positional effects of inserting CPE and PAS along the 3′ UTR. Plotted are mean differences in

predicted tail-length change. Drop near position 17 reflects PAS disruption (Supplementary Fig. 3e). Shaded areas, standard error. Inset, last 100 nt of the 3′ UTR. **f** Predicted effects of CPE-PAS spacing. Mean differences in PAL-AI-predicted tail-length change conferred upon inserting a CPE in silico are plotted as a function of the relative distance between CPE and PAS. Shaded areas, standard error. The gray box, CPE-PAS overlapping positions. **g** PAL-AI-predicted effects of single-nucleotide substitutions in the *mos.L* 3′ UTR. The heatmap indicates the difference in predicted tail-length change (DPTLC) for each substitution (x, original; y, alternative). Line plots indicate max (red) and min (blue) mutational outcomes at each position. The logo plot indicates the importance of each nucleotide, with the height normalized to the negative value of the average outcome of three possible substitutions. Dashed rectangles, CPE and PAS. The arrow points to an instance of a new CPE, generated by a G-to-U substitution, and its associated increase in tail-length change. **h** PAL-AI-predicted effects of single-nucleotide substitutions in the *tpx2.L* 3′ UTR, plotted as in (**g**). The solid arrow points to an example of a substitution to a more optimal PAS-flanking nucleotide; the dashed arrows point to optimal CPE-flanking nucleotides[14].

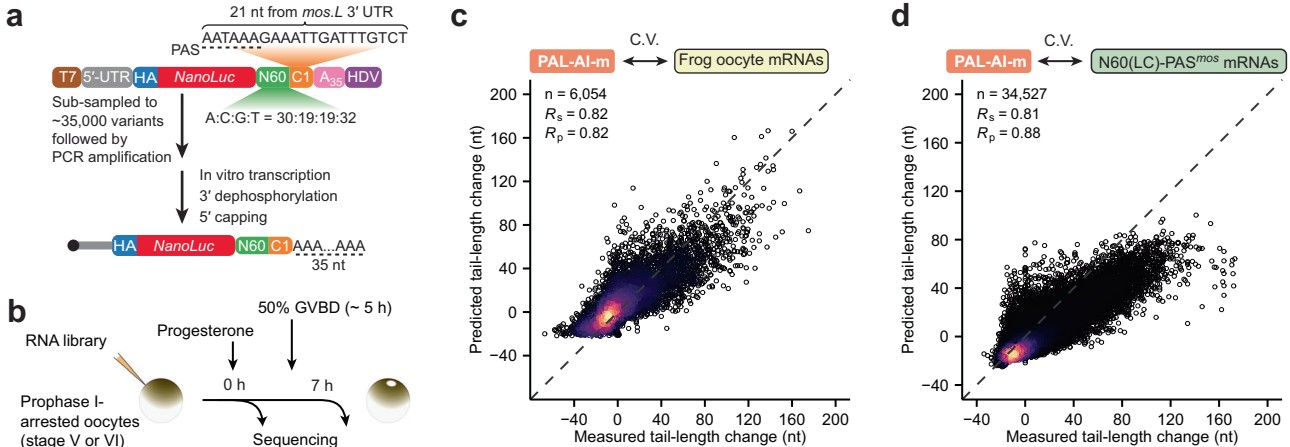

**Fig. 3 | PAL-AI predicts tail-length changes of synthetic mRNAs in frog oocytes. a** Schematic of the N60(LC)-PAS*mos* mRNA library used for injection. **b** Experimental scheme for mRNA library injection and sample collection. **c** Performance of the PAL-AI-m model in predicting tail-length changes of frog endogenous mRNAs.

Otherwise, this panel is as in Fig. 1c. **d** Performance of the PAL-AI-m model in predicting tail-length changes of mRNAs in the N60(LC)- PAS*mos* library. Otherwise, this is as in Fig. 1c.

to any PAS variant reduced the predicted tail-length increase by 7.7–11.7 nt (Supplementary Fig. 3g). Additionally, in the CPE*mos*-N60 reporter 3′ UTR library[14], when variants containing either an AAUAAA or AUUAAA were excluded, none of the remaining 6-mers were associated with tail lengthening (Supplementary Fig. 3h). Together, these analyses indicate that although non-canonical PAS variants can direct mRNA cleavage and polyadenylation[42], they do not facilitate mRNA polyadenylation in the cytoplasm—at least not during oocyte maturation.

To gain more insights into properties that PAL-AI learned at the transcript level, we examined predicted consequences of all possible point substitutions at the last 100 nt of the *mos.L* mRNA. In silico mutations that perturbed either the PAS at position −21 or the CPE at position −86 caused considerable decreases in predicted tail lengthening, whereas a G-to-U mutation at position −62 created a new CPE, thereby causing an increase in predicted tail lengthening (Fig. 2g). *mos.L* also contains a second CPE falling near the PAS, at position −27. This PAS-proximal CPE was expected to be suboptimal for cytoplasmic polyadenylation because its separation from the PAS of only 1 nt was too short[14]. Notably, PAL-AI recognized this proximity feature and correctly predicted only mild mutational outcomes when this motif was disrupted (Fig. 2g).

In another example, the *tpx2.L* mRNA, PAL-AI identified two CPEs predicted to be important for tail lengthening. Furthermore, PAL-AI predicted decreases in tail lengthening when Gs at positions −66 and −36 (both at −2 positions relative to CPEs) were mutated and an increase in tail-length change for a C-to-G substitution at position −12 (+1 relative to the PAS) (Fig. 2h)—all observations consistent with the contextual features of the CPE and the PAS identified from analyses of millions of reporters[14]. Similar results were observed for other examples of known substrates of cytoplasmic polyadenylation (Supplementary Fig. 3i, j).

Together, these analyses indicated that PAL-AI successfully captured the sequence elements and their contextual features important for promoting cytoplasmic polyadenylation, thereby enabling it to accurately predict poly(A) tail-length changes of frog mRNAs during oocyte maturation.

**PAL-AI predicts poly(A) tail-length change of injected mRNAs**
Next, we asked if PAL-AI was able to predict tail-length changes of injected reporter mRNAs. To this end, we amplified a small subset of a 3′-UTR reporter library (N60-PAS*mos*)[14] to generate a library with only ~35,000 variants (Fig. 3a), which we called N60(LC)-PAS*mos*. The

reduced sequence complexity of this library enabled acquisition of tail-length distributions of individual mRNA sequences. We injected this mRNA library into prophase I-arrested frog oocytes, collected RNA samples at 0 h and 7 h post-progesterone treatment, and used high-throughput sequencing[14] to identify the 3′ UTR variant and determine its tail-length distribution at each of the two stages of maturation (Fig. 3b).

When using tail-length changes measured for these injected mRNAs to test our model trained on endogenous mRNAs, our model performed well (Supplementary Fig. 4a). Indeed, correlation coefficients observed when testing on the injected mRNAs ($R_s = 0.77$, $R_p = 0.83$) resembled those observed when testing on endogenous mRNAs (Fig. 1e, $R_s = 0.82$, $R_p = 0.82$). However, the model appeared to under-predict the magnitude of the tail-lengthening effects (Supplementary Fig. 4a). A few possibilities could explain the systematic difference in the magnitude of tail-lengthening observed for injected mRNAs compared to that predicted by PAL-AI. First, the sequence lengths of the training and the testing datasets were markedly different. Endogenous mRNAs used for training PAL-AI had few 3′ UTRs shorter than 100 nt (420 out of 6054), whereas 3′ UTRs of the injected mRNAs were all relatively short (81 nt). Second, due to experimental variability, particularly the variable time it takes for oocytes of different batches to mature, the endogenous mRNAs and injected mRNAs may have been collected at somewhat different maturation stages. Third, the 3′ UTRs of the reporter mRNAs consisted of arbitrary sequences, whose composition may differ substantially from natural mRNA sequences.

To test whether sequence composition contributed to the discrepancy, we modified PAL-AI's last layer to enable dataset-specific multi-output predictions (Supplementary Fig. 4b). Training on alternating batches of endogenous and injected mRNAs maintained performance on endogenous mRNAs (Fig. 3c, $R_s = 0.82$, $R_p = 0.82$) while enhancing performance on injected mRNAs (Fig. 3d, $R_s = 0.81$, $R_p = 0.88$). To distinguish this multi-output model from the original PAL-AI model, we termed this model as PAL-AI-m ("m" for multi-output) and the original model as PAL-AI-s ("s" for single-output).

Despite the improvement, the PAL-AI-m model still under-predicted tail lengthening for most long-tailed (≥80 nt) mRNAs from the injected library. When we re-trained the model with measurements from only injected mRNAs, PAL-AI performed significantly better when tested on held-out data (Supplementary Fig. 4c, $R_s = 0.86$, $R_p = 0.91$). In silico mutagenesis analyses indicated that the new model also learned the expected sequence elements important for tail lengthening and

their contextual features (Supplementary Fig. 4d–f). This training approach on synthetic, nonbiological reporter sequences also ensured that these learned elements and features are causative for tail-length changes, rather than correlated to causative features.

## Predicted effects of single-nucleotide substitutions correspond with experimentally determined effects

The ability of PAL-AI to predict the fate of injected mRNAs enabled direct assessment of its ability to predict the consequences of mutations, including all possible single-nucleotide substitutions. To this end, we generated a single-nucleotide mutagenesis library of reporter mRNAs with 3′-UTR sequences derived from the last 100 nt of 3′ UTRs of ten mRNAs selected from frog oocytes. For each of the ten 100-nt UTR fragments, each nucleotide of the 3′ UTR was substituted with one of the three alternative nucleotides, resulting in 3000 variants (Fig. 4a). These 3000 variants, together with the non-substituted wild-type molecules, were injected into oocytes and the effect of each substitution on tail-length changes was analyzed over the course of maturation. Of the ten 3′-UTR fragments, eight (*atp1a1.S*, *ccnb1.2.L*, *ccnb2.L*, *lima1.L*, *mad2l1.L*, *magoh.S*, *mos.L*, and *tpx2.L*) directed cytoplasmic polyadenylation upon injection into oocytes, whereas two (*aurkaip1.L* and *dbf4.L*) directed no tail lengthening during frog oocyte maturation[14].

When comparing the predicted difference in tail-length change to that measured between each mutant and the wild-type, the PAL-AI-m model performed significantly better compared to the PAL-AI-s model for eight out of ten mRNAs (Fig. 4b, Supplementary Data 5). This improved performance suggested that the PAL-AI-m model successfully learned additional features from the non-natural sequences of the N60(LC)-PAS$^{mos}$ library. The PAL-AI-m model achieved high accuracy ($R_p$: 0.78–0.90) for six mRNAs (*atp1a1.S*, *lima1.L*, *mad2l1.L*, *magoh.S*, *mos.L*, and *tpx2.L*; Figs. 4b, d–g, 5a, b, Supplementary Fig. 5a–f) and moderate accuracy ($R_p$: 0.58–0.62) for three mRNAs (*aurkaip1.L*, *ccnb1.2.L*, and *ccnb2.L*; Fig. 4b, Supplementary Figs. 6a, b, d, e, 7a, b).

Concordance between predicted and experimental outcomes was particularly clear for substitutions that disrupted either a CPE or PAS (Figs. 4c, f, g, 5a, Supplementary Fig. 5d–f). Substituting the canonical PAS (AAUAAA) with the alternative variant AUUAAA severely affected tail lengthening, but not as severely as observed for any of the other single-nucleotide substitutions of AAUAAA (Fig. 4c). These experimental findings validated our in silico mutagenesis conclusion that noncanonical PAS variants do not support cytoplasmic polyadenylation during oocyte maturation.

At the individual mRNA level, ablation of any of the three motifs (two CPEs and one PAS) within *mos.L* mRNA led to substantial decreases in tail lengthening, as predicted by PAL-AI-m (Fig. 4f). The predicted weaker contribution to cytoplasmic polyadenylation by the CPE proximal to the PAS compared to that by the CPE further upstream was also validated (Fig. 4f). Similar results illustrating PAL-AI-m's ability to predict the importance of these motifs and their contextual features were also observed for other mRNAs (Fig. 4g, Supplementary Fig. 5). Notably, the predicted differences in tail-length changes observed between many variants and the wild-type were smaller than those measured, perhaps due to the same reasons previously put forward to explain the under-prediction of changes for the injected N60(LC)-PAS$^{mos}$ library (Supplementary Fig. 4a).

In another example, *mad2l1.L* mRNA had no CPE within the last 100 nt of its 3′ UTR but contained seven UGU/UGUUU motifs (Fig. 5a). Mutations within these motifs caused modest but consistent reductions in tail lengthening, agreeing with their detectable but weak roles in cytoplasmic polyadenylation. In contrast, substitutions that introduced a single CPE led to large increases in tail lengthening, almost all of which were correctly predicted by PAL-AI-m (Fig. 5a, b).

Despite achieving good performance for the six aforementioned mRNAs ($R_p$ = 0.78–0.9), PAL-AI-m performed only moderately well on

three mRNAs (*aurkaip1.L*, *ccnb2.L*, and *ccnb1.2.L*, $R_p$ = 0.58, 0.62, and 0.59, Supplementary Figs. 6b, e, 7a) and did not perform well at all on one mRNA (*dbf4.L*, $R_p$ = 0.21, Fig.5c). The wide-type *dbf4.L* mRNA, which had no CPE and a relatively weak PAS (AUUAAA), had little tail lengthening, and thus most single-nucleotide substitutions had negligible opportunity to reduce tail lengthening (Fig. 5d), which largely explained their relatively weak effects for this mRNA. PAL-AI-m did predict moderate increases in tail lengthening for a few substitutions that would create CPEs predicted to lengthen tails, but by no more than 10 nt (Fig. 5c). Although some of these predicted increases were validated (positions −61 and −51), others, particularly those downstream the PAS (positions −35, −31, −7 and −6) were not (Fig. 5d). To understand these discrepancies, we predicted the secondary structure of the last 100 nt of the wild-type *dbf4.L* 3′ UTR using EternaFold[43]. The substitutions at positions −61 and −51 both created CPEs in loop regions, whereas the substitutions at positions −35, −31, −7 and −6 all created CPEs with nucleotides paired in a stem loop (Fig. 5d, e), which likely prevented these CPEs from being recognized by the CPEB1 protein.

For *aurkaip1.L* and *ccnb2.L* mRNAs, PAL-AI-m had difficulty predicting increased tail lengthening caused by substitutions in consecutive positions within a few UTR segments (*aurkaip1.L*: positions −34 to −30 and −19 to −15, Supplementary Fig. 6a; *ccnb2.L*: positions −60 to −55 and −15 to −10, Supplementary Fig. 6d). These substitutions appeared to destabilize stem loops that were sequestering PAS elements in the wild-type UTRs (Supplementary Fig. 6c, f), thus making these elements more accessible for CPSF binding. In support of this hypothesis, substitutions that further stabilized these stem loops reduced tail lengthening (*aurkaip1.L*: A-to-G at position −35 and C-to-U at position −14, Supplementary Fig. 6a, c; *ccnb2.L*: C-to-G at position −54 and C-to-G at position −16, Supplementary Fig. 6d, f). Together, these analyses indicated that sequences within the 3′ UTR can indirectly affect poly(A) tail-length changes by modulating the structural accessibility of the CPE and the PAS, a mechanism that likely applies to select mRNAs[30] and thus had not yet been effectively learned by PAL-AI-m.

As for the *ccnb1.2.L* mRNA, PAL-AI-m under-predicted the effects of a CPE that overlapped with the PAS (Supplementary Fig. 7a, b). Binding of CPEB1 to this overlapping CPE presumably prevented CPSF from recognizing the PAS, thus inhibiting cytoplasmic polyadenylation as previously suggested[44]. Supporting this hypothesis, the C-to-U mutation at position −18 of *magoh.S* mRNA created an overlapping CPE, which reduced tail lengthening (Supplementary Fig. 5f).

Despite some discrepancies between the predictions and experimental results, these analyses demonstrated that, overall, PAL-AI can accurately predict tail-length changes caused by single-nucleotide mutations within mRNA 3′ UTRs.

## PAL-AI predicts poly(A) tail-length change in mammalian oocytes

Because principles of poly(A) tail-length control during oocyte maturation are highly conserved among frogs, mice, and humans[14], we asked if PAL-AI trained from frog oocyte data could predict tail-length changes during mammalian oocyte maturation. For mouse oocytes, PAL-AI achieved moderate accuracy (Supplementary Fig. 8a; $R_s$ = 0.57, $R_p$ = 0.58). The performance of the model was likely affected by unmatched developmental stages between frogs and mice, and also global deadenylation that was independent of 3′-UTR sequences[13,14]. Similar predictive power for PAL-AI was also observed for human oocytes using published datasets (Supplementary Fig. 8b; $R_s$ = 0.61, $R_p$ = 0.59), even though the human measurements were acquired using a different sequencing protocol and platform (PacBio)[24].

To minimize prediction errors arising from species-specific sequence composition and experimental variability, we developed a multi-species PAL-AI model trained on pooled data from frogs, mice,

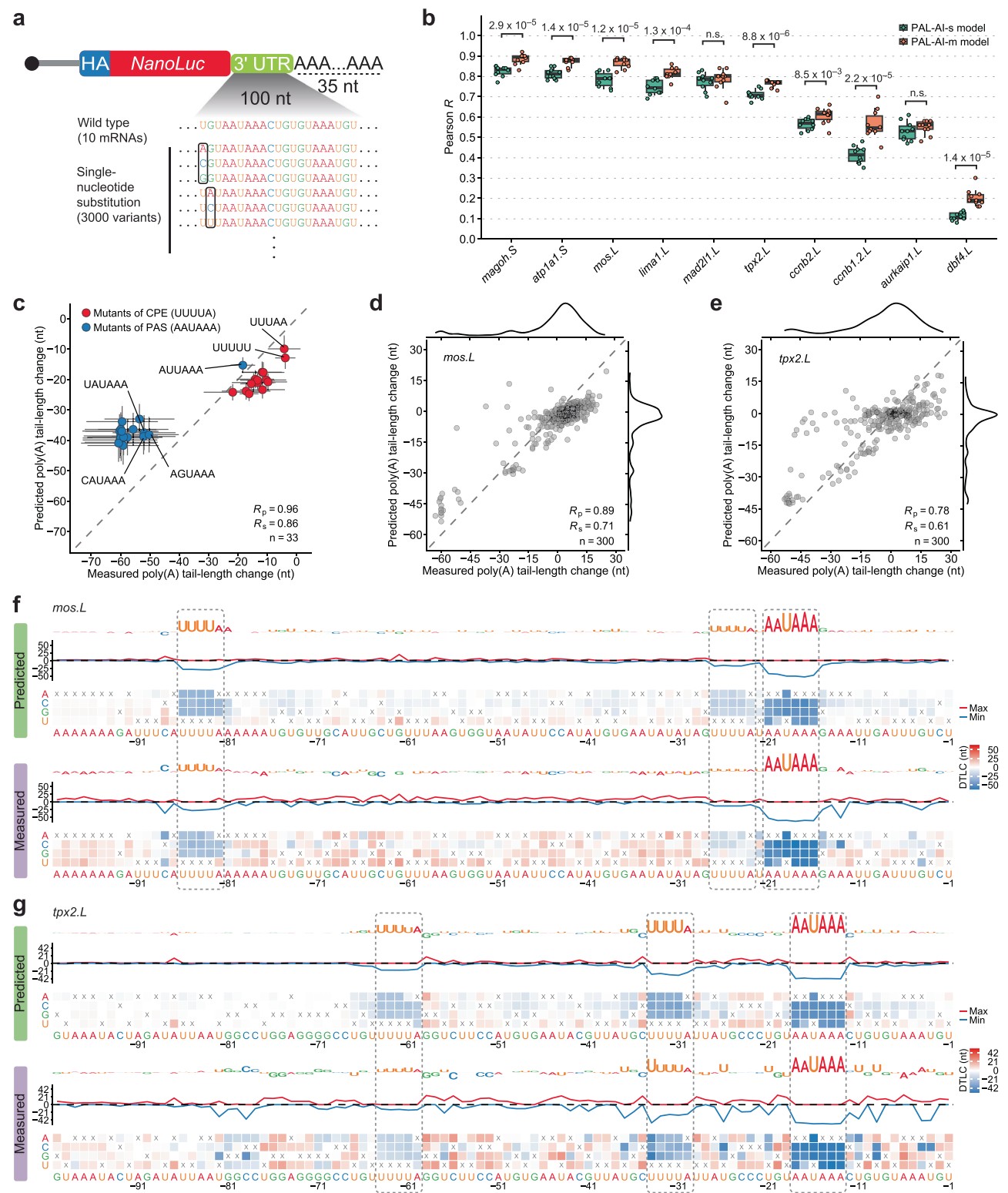

and humans, using the same approach as for our frog endogenous mRNAs and injected mRNAs. This model, termed PAL-AI-ms ("ms" for multi-species), showed improved performance for both mouse (Fig. 6a; $R_s = 0.63$, $R_p = 0.64$) and human oocytes (Fig. 6b; $R_s = 0.68$, $R_p = 0.66$), while maintaining comparable accuracy for frog oocytes (Supplementary Fig. 8c; $R_s = 0.79$, $R_p = 0.77$). In addition, the model's predictions also correlated well with independent tail-length measurements obtained using Oxford Nanopore Technologies (ONT) for mRNAs during mouse oocyte maturation[13] (Supplementary Fig. 8d;

$R_s = 0.69$, $R_p = 0.69$). Although ONT data showed larger absolute tail-length changes compared to our measurements obtained using Illumina sequencing (HiSeq 2000), both datasets exhibited consistent trends (Supplementary Fig. 8e). This systematic difference in magnitude likely explained why the model under-predicted tail-length changes in the ONT data (Supplementary Fig. 8d).

In frog, mouse, and human oocytes, tail-length changes strongly correlate with translational efficiency changes[10,13,14]. As a result, our model-predicted tail-length changes were expected to be informative

**Fig. 4 | Experiments validate predicted effects of single-nucleotide substitutions. a** Schematic of the library of single-nucleotide substitutions used for injection. **b** Summary of the relationships between the measured and the PAL-AI-s- or PAL-AI-m-predicted effects of single-nucleotide substitutions. For each of the 10 mRNAs in the single-nucleotide mutagenesis library, Pearson $R$ values were calculated based on the differences in tail-length change predicted by the indicated model (key) and those measured for each single-nucleotide substitution, comparing between 0 h and 7 h post-progesterone treatment. The top-performing model trained from each fold of the ten-fold cross-validation was used for predictions (ten models total for each architecture). $P$ values are from one-sided t-tests. Box and whiskers indicate the 10th, 25th, 50th, 75th, and 90th percentiles. n.s. not significant ($P \geq 0.01$). **c** PAL-AI-m prediction of the effects of single-nucleotide substitutions that disrupt either a CPE or a PAS (AAUAAA). Error bars indicate the standard error of the mean of variants from the single-nucleotide substitution library bearing the indicated substitutions. **d** PAL-AI-m prediction of the effects of

single-nucleotide substitutions of the *mos.L* mRNA. Plotted for each single-nucleotide substitution is the difference in tail-length change predicted by PAL-AI-m compared to that measured for that mRNA variant injected into frog oocytes and collected 0 h and 7 h post-progesterone treatment. On the sides are density distributions for the predicted and measured differences in tail-length change. **e** PAL-AI-m prediction of the effects of single-nucleotide substitutions of the *tpx2.L* mRNA; otherwise, as in (**d**). **f** Impact of single-nucleotide substitutions on predicted and measured tail-length changes for the *mos.L* mRNA. Shown are results for the last 100 nt of the *mos.L* 3′ UTR. The heatmaps indicate the predicted (top) and measured (bottom) differences in tail-length change (DTLC) as a result of changing each nucleotide (x axis) to each of the three non-wild-type alternatives (y axis), with x indicating the non-substituted, wild-type nucleotide. The line plots and logo plots are as in Fig. 2g. Dashed rectangles indicate the CPE and PAS. **g** Impact of single-nucleotide substitutions on predicted and measured tail-length changes for the *tpx2.L* mRNA; otherwise as in (**f**).

regarding up- and down-regulation of mRNA translation—not only for endogenous mRNAs but also for any mRNAs with known 3′-UTR sequences. To confirm this expectation, we predicted tail-length changes for a series of reporter mRNAs with different 3′-UTR sequences used for assessing translational regulation during mouse oocyte maturation[12]. Our model-predicted tail-length changes corresponded well with the reported translational efficiency changes (Fig. 6c; $R_s = 0.89$, $R_p = 0.91$).

Although tail-length control during oocyte maturation is highly conserved among vertebrates, tail lengthening and associated translational upregulation are known to be required for mRNAs from only a few genes. Conserved tail lengthening of mRNAs from orthologous genes may shed light on the functional significance of tail lengthening during oocyte maturation and early development, but mRNAs from only 19 genes with conserved tail lengthening have been identified[14]. Part of the reason for this small number is that most mRNAs could not be evaluated because they lacked tail-length measurements in at least one of the three species[14]. To overcome this limitation, we used our model-predicted tail-length changes for cases in which tail-length measurements were unavailable, resulting in the identification of another 264 genes (57, 108, and 99 with one, two, and three predicted values, respectively) whose mRNA poly(A) tails were either extended or predicted to be extended ≥15 nt in all three species (Fig. 6d, Supplementary Fig. 8f, g). Most of these mRNAs had increased translational efficiency, and for those for which translational efficiency changes had not been measured, PAL-AI-predicted tail-length increases could be informative. A Gene Ontology analysis of these genes revealed enrichment of molecular pathways such as cell cycle, microtubule cytoskeleton organization, cell division, and chromosome organization, which are crucial for meiosis and presumably prepare the zygote for cell divisions of early embryonic development (Fig. 6e).

**Variants predicted to disrupt poly(A)-tail lengthening are under negative selection in humans and throughout vertebrates**

Next, we asked if genetic variants predicted to influence tail-length changes during oocyte maturation have been under evolutionary selection. We analyzed human 3′ UTRs by using PAL-AI-ms to predict the effects of all possible single-nucleotide mutations within their last 100 nt. For each position, we calculated the average difference in predicted tail-length change when mutated to alternative nucleotides and classified the wild-type nucleotide by the direction of the effect and severity when mutated: tail-length reduction (severe [≤ −25 nt], moderate [−25 to −15 nt], mild [−15 to −5 nt]), minimal-impact [−5 to 5 nt], and tail-length increase (mild [5 to 15 nt], moderate [15 to 25 nt], severe [>25 nt]). Comparing these predictions to phyloP scores, a conservation metric based on the alignment of 100 vertebrate genomic sequences[45,46], revealed strong evolutionary constraints, with conservation levels tracking predicted functional impact. For example, nucleotides for which mutations would cause severe tail-length reduction showed

significantly higher phyloP scores than moderate-impact sites (Mann–Whitney U test), which in turn were more conserved than mild-impact sites, with minimal-impact sites being least conserved (Fig. 7a, Supplementary Fig. 9a, Supplementary Data 5). Nucleotides for which mutations would cause tail-length increase showed a parallel but weaker trend, with only mild/moderate-impact sites exhibiting significantly elevated conservation relative to minimal-impact sites (Fig. 7a, Supplementary Fig. 9a, Supplementary Data 5). This is likely in part because tail-length shortening during oocyte maturation is non-specific[13,14,22,23] and therefore very few sites, when mutated, would result in tail-length increase (Supplementary Data 5). Overall, these findings indicate that genomic positions at which mutations would substantially alter poly(A) tail-length changes during oocyte maturation have been under purifying selection in the vertebrate lineage.

We also asked if genetic variants predicted to influence tail-length control during oocyte maturation might have functional consequences for human health. To this end, we analyzed ~8.4 million 3′-UTR variants from the All of Us Research Program[47] using PAL-AI-ms. For each variant, we calculated its predicted effect size as the difference in predicted tail-length change relative to the reference allele (Fig. 7b). Focusing on the last 100 nt of each 3′ UTR (~0.85 million variants), we binned variants by predicted effect size the same way as we had done for the conservation analysis and examined their relative fractions in each bin for groups with different allele frequencies. This analysis revealed a striking allele-frequency-dependent depletion of tail-lengthening-disrupting variants. For example, variants predicted to severely disrupt tail-lengthening showed significant depletion (2-, 3-, 7- and 10-fold) in common variants (allele frequencies >0.01%, >0.1%, >1%, and >10%, respectively) compared to singletons ($P = 1.7 \times 10^{-10}$, $6.8 \times 10^{-8}$, $8.3 \times 10^{-7}$, and $8.1 \times 10^{-4}$, Fisher's exact test, Supplementary Data 5). Depletion of frequent alleles was also significant, albeit to a lesser degree, for variants predicted to impart moderate (1.5–1.9 fold) and mild (1.3–1.7 fold) reduced tail-lengthening (Fig. 7c, Supplementary Data 5). This depletion pattern remained significant even after excluding all variants that would introduce or ablate PAS motifs (Supplementary Fig. 9b, Supplementary Data 5), which were presumably also under negative selection due to their nuclear role in specifying mRNA cleavage and polyadenylation[48]. In contrast, gain-of-function variants were not consistently depleted in common alleles compared to singleton variants (Fig. 7c, Supplementary Data 5). Similar results were observed when PAL-AI was applied to the gnomAD v4.1 datasets[49] (Fig. 7d, Supplementary Fig. 9c, d, Supplementary Data 5). Together, these analyses indicated that variants that disrupt poly(A) tail-length control, particularly those that disrupt tail lengthening during oocyte maturation, are under negative selection in humans.

## Discussion

We developed PAL-AI, a neural network model that predicted tail-length changes in frog, mouse, and human oocytes using only sequence information. PAL-AI captured consequential sequence

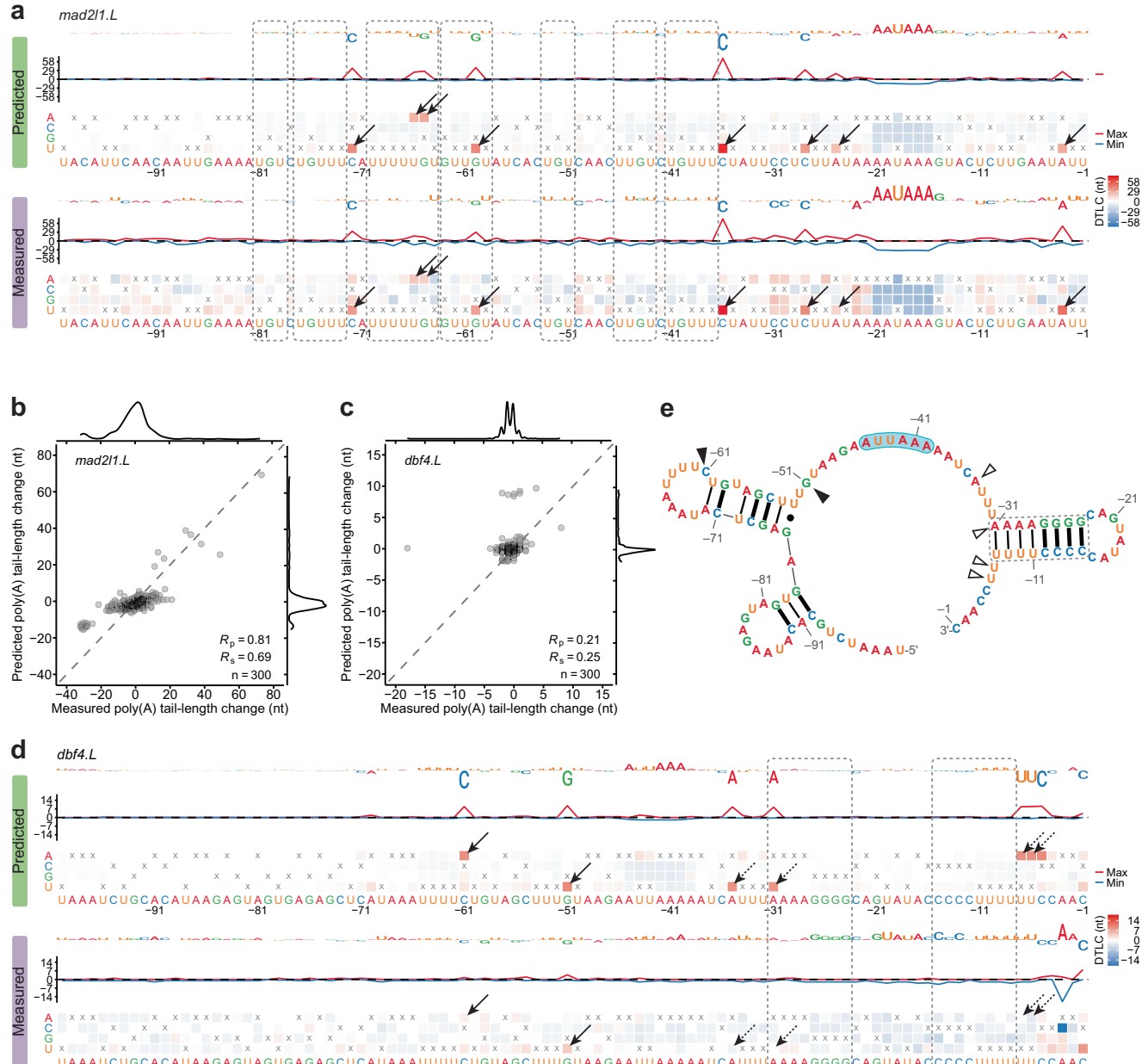

**Fig. 5 | Auxiliary motifs and structural accessibility influence tail-length changes. a** Impact of single-nucleotide substitutions on predicted and measured tail-length changes for the *mad2l1.L* mRNA. Dashed rectangles indicate UGU/UGUUU elements that each appears to modestly promote poly(A)-tail lengthening. The arrows point to instances of new CPEs generated by substitutions and their associated increases in tail-length changes. Otherwise, this panel is as in Fig. 4f. **b** PAL-AI-m prediction of the effects of single-nucleotide substitutions of the *mad2l1.L* mRNA. Otherwise, this panel is as in Fig. 4d. **c** PAL-AI-m prediction of the effects of single-nucleotide substitutions of the *dbf4.L* mRNA. Otherwise, this panel is as in Fig. 4d. **d** Impact of single-nucleotide substitutions on predicted and measured tail-length changes for the *dbf4.L* mRNA. Dashed rectangles indicate two regions predicted to pair with each other, forming the stem of a hairpin. Arrows indicate CPE-creating substitutions with tail-length changes either correctly (solid) or incorrectly (dashed) predicted by PAL-AI-m. **e** EternaFold-predicted maximum expected accuracy secondary structure of the last 100 nt of the *dbf4.L* 3′ UTR. The dashed rectangle indicates the stem of the hairpin highlighted in (**d**). Triangles indicate positions where CPE-creating substitutions with tail-length changes either correctly (solid) or incorrectly (hollow) predicted by PAL-AI-m, as shown in (**d**). The PAS is shaded light blue. Base pairings are indicated with thick lines for G–C, thin lines for A–U, and a dot for the G–U wobble.

motifs and contextual effects without any feature generation, thereby allowing us to predict tail-length changes of mRNAs with any sequence. Moreover, PAL-AI helped us identify UGU/GUU-containing motifs contributing to poly(A)-tail lengthening. Although previously implicated in regulating translation[35,38,39,50–52], these motifs had not been associated with tail-length control. Our analyses of the injected mRNA reporter libraries further demonstrated that their tail-lengthening effects were causal rather than correlative with other features of endogenous mRNAs. Individually, these motifs contributed

weakly to polyadenylation, but multiple copies did result in considerable tail lengthening, as illustrated by the *mad2l1.L* mRNA. Their precise roles in affecting tail lengths might be context-dependent, as some might serve as CPE-like motifs promiscuously bound by CPEB1, whereas others might be recognized by RNA-binding proteins such as DAZL, which also interacts with CPEB1 and thus might help facilitate recruiting CPEB1 to the 3′ UTR.

The ability of PAL-AI to predict tail-length changes of both endogenous mRNAs and injected mRNAs enabled us to capture the

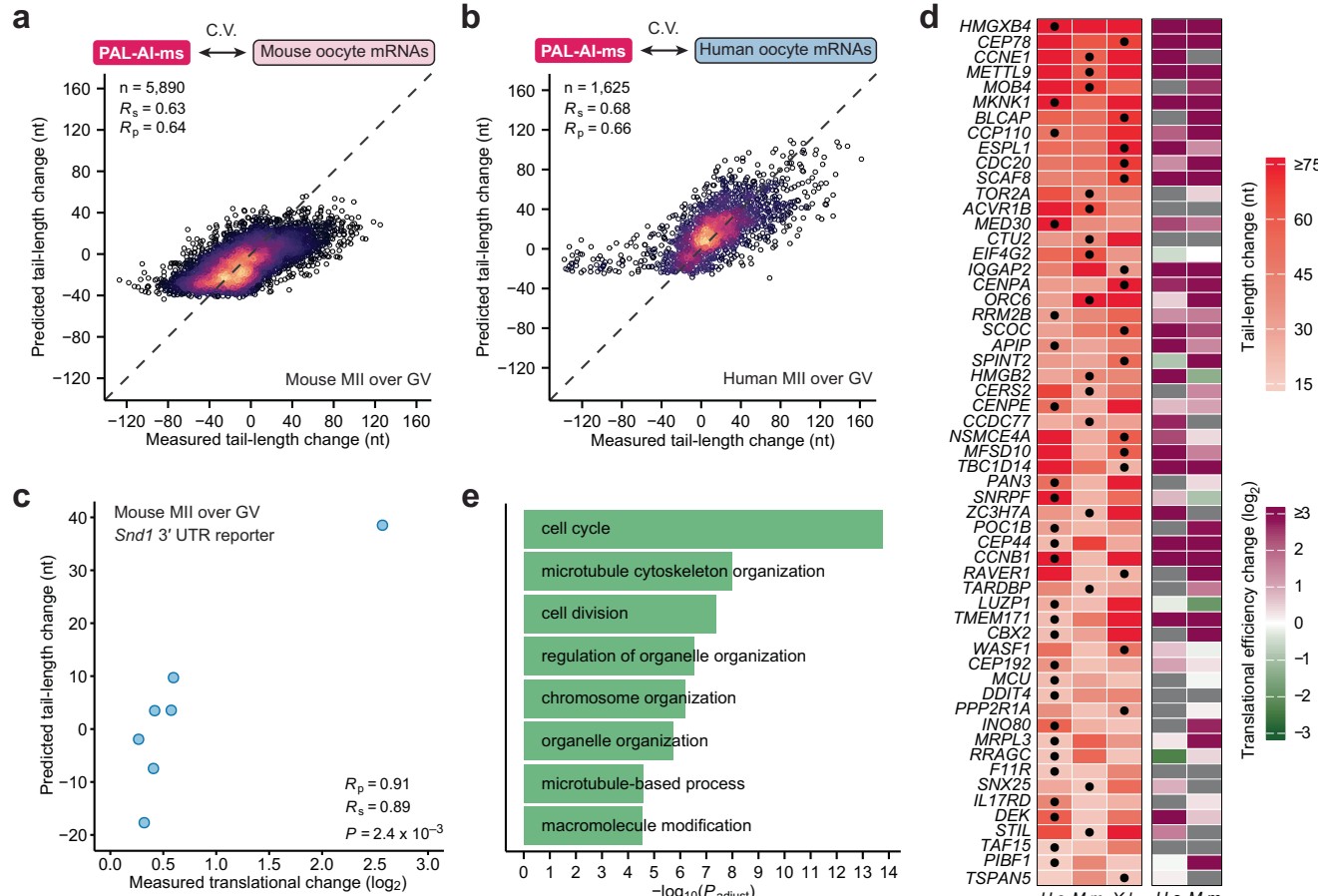

**Fig. 6 | Multi-species PAL-AI predicts tail-length changes of endogenous mRNAs of mouse and human oocytes. a** Performance of the multi-species PAL-AI model on mouse mRNAs. Plotted are the tail-length changes predicted by PAL-AI-ms to occur upon mouse GV-to-MII oocyte maturation as a function of the measured changes. Each point represents a unique poly(A) site of an endogenous mRNA. Otherwise, this panel is as in Fig. 1c. **b** Performance of the multi-species PAL-AI model on human mRNAs. Otherwise, this panel is as in (**a**). **c** Ability of predicted tail-length changes to explain translational efficiency changes. Plotted is the relationship between the multi-species PAL-AI model-predicted tail-length changes and translational efficiency changes measured for seven mRNA reporters upon mouse GV-to-MII oocyte maturation[12]. The *P* value was from a one-sided correlation test.

**d** Genes predicted or shown to have substantial mRNA tail-lengthening (≥15 nt) in human (*H.s.*), mouse (*M.m.*), and frog (*X.l.*) oocytes. Heatmaps show measured or predicted (dots) tail-length changes (left) and measured translational efficiency changes (right), comparing between human GV and MII oocytes, mouse GV and MII oocytes, and frog oocytes 0 h and 7 h post-progesterone treatment. Only genes with the model-predicted values in one of the three species were shown. Gray indicates values not available. **e** Select ontologies enriched for genes with substantial mRNA tail-lengthening (≥15 nt, predicted, if not measured) in human, mouse, and frog oocytes. *P* values were from one-sided Fisher's exact tests and corrected for multiple testing[72].

consequences of most single-nucleotide substitutions. For a few mRNAs, PAL-AI under-performed in predicting the outcomes of substitutions—often those substitutions that appeared to alter the structural accessibility of CPE or PAS elements. However, when we combined base-pairing probability, as predicted by RNAfold[32], with mRNA sequences as the input for our model, the performance did not significantly improve, which suggested that more accurate parameterization of mRNA structure would be needed for the model to account for structural accessibility. Perhaps incorporating data from in-oocyte structural probing of mRNAs will further improve the model predictions[53–56].

Because of the strong influence of tail length on translational efficiency in oocytes, we envision that our model can be used to engineer mRNA 3'-UTR sequences for desired protein expression in these systems. Furthermore, by predicting tail-length changes that had not yet been measured in frog, mouse, or human oocytes, our model helped identify a group of genes that appear to undergo conserved mRNA tail-lengthening during oocyte maturation. Although for mRNAs of most of these genes, the importance of translational upregulation during oocyte maturation and early embryogenesis still needs to be examined, knowledge of these genes can inform efforts to identify mutations that affect human female fertility.

Indeed, genetic variants predicted by PAL-AI to disrupt tail-length control during oocyte maturation have been under negative selection in vertebrates including humans. Presumably, some of these variants perturb sequence elements or contextual features important for cytoplasmic polyadenylation, thus leading to failed translational activation of mRNAs encoding proteins important for oocyte maturation or early embryonic development. While not evidently observed in this study, as larger genomic data sets become available and more sensitive analyses are conducted, negative selection might also be detected for mutations predicted to lead to inappropriate activation of translationally repressed mRNAs during these developmental transitions.

Although the observed depletion of variants predicted to disrupt tail lengthening persisted after excluding PAS-perturbing substitutions, we cannot rule out that some of this selection pressure may reflect pleiotropic effects on other RNA-processing events. For instance, U-rich sequences resembling the CPE near the mRNA 3' end can serve as binding platforms for Fip1 in nuclear pre-mRNA processing[57,58]. Variants disrupting these motifs could simultaneously

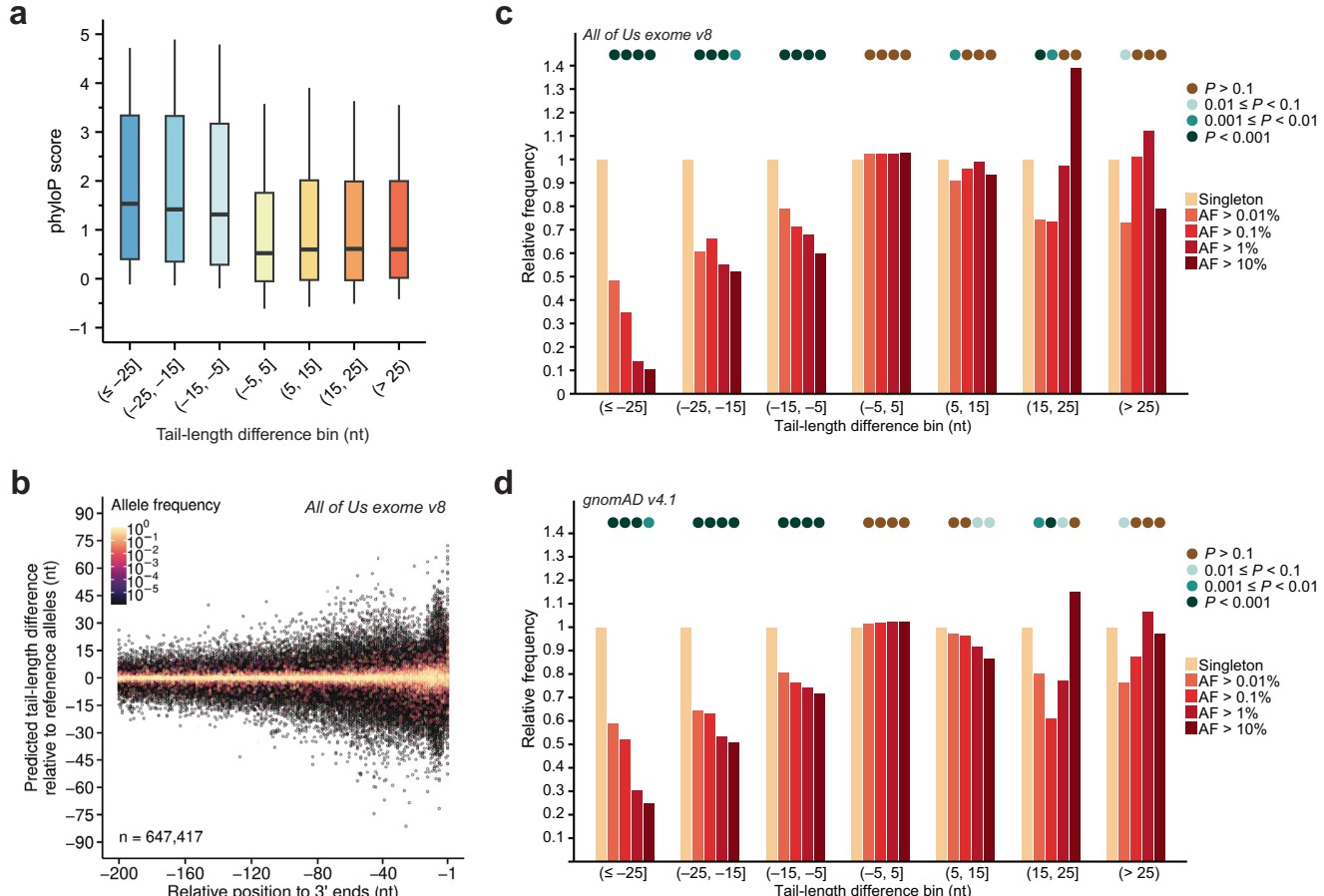

**Fig. 7 | Variants predicted to disrupt poly(A)-tail lengthening are selected against in the human population and throughout vertebrates. a** Negative selection among sequenced vertebrate species of variants predicted to be more disruptive of tail lengthening. Shown are distributions of phyloP scores[45,46], grouped by the averaged differences in PAL-AI-predicted tail-length changes for individual nucleotides when substituted to each of the three alternative nucleotides. Only positions within the last 100 nt of 3′ UTRs were included in this analysis. For the tail-length difference bin, parentheses indicate values not included, while square brackets indicate values included. Box and whiskers indicate the 10th, 25th, 50th, 75th, and 90th percentiles. Statistical test results are reported in Supplementary Fig. 9a and Supplementary Data 5. **b** Predicted effects on the tail-length change for human variants. Shown is the PAL-AI-predicted difference in the tail-

length change for each variant reported in the All of Us Research Program (exome callset v8) and that of the reference allele, plotted as a function of the variant position in its 3′ UTR. Colors indicate allele frequencies (key). **c** Depletion in human 3′ UTRs of alleles predicted to disrupt poly(A)-tail lengthening. Shown are relative frequencies of variants, grouped by differences in predicted tail-length change, among cohorts of variants with different allele frequencies (AF) reported in the All of Us Research Program (exome callset v8). Only variants within the last 100 nt of 3′ UTRs were included in this analysis. Binned *P* values (circles at the top) were from one-sided Fisher's exact tests performed for each cohort against singletons. Statistical test results are reported in Supplementary Data 5. **d** Same as (**c**), but for variants reported in gnomAD v4.1 dataset.

impair both cytoplasmic polyadenylation and nuclear 3′-end formation. Future integration of PAL-AI with tools capable of predicting 3′-end-processing defects, such as APARENT2[48] or PolyaID[59], could help distinguish these potentially confounding selective pressures.

Past efforts to identify genetic causes of female infertility have primarily focused on variants in coding regions, perhaps in part due to the use of exome sequencing[19]. However, mutations that explain defects in oocytes and early embryos associated with female infertility fall in fewer than 20 genes and are found in only a small portion of patients[60]. Given the importance of tail-length and translational control in human oocytes and early embryos, as well as our analyses of variant abundancies in the human population, some unexplained cases are presumably due to disruptive mutations occurring in 3′ UTRs. With more genomes with female reproductive failures being sequenced, we will be able to start assessing contributions by these variants, and PAL-AI can be a valuable tool complementing current approaches, such as genome-wide association studies, to help nominate causal variants within 3′ UTRs.

## Methods

### Annotations of poly(A) sites and 3′ UTRs
Annotations of poly(A) sites and 3′ UTRs of frog, mouse, and human oocytes were obtained from a previous study[14].

### Linear regression models for poly(A) tail-length changes
Poly(A) tail-length changes for each isoform of frog oocyte endogenous mRNAs were calculated as the difference between median tail lengths measured at 0 h and 7 h post-progesterone treatment, requiring at least 50 poly(A) tags at both time points, as reported previously[14]. Only 3′ UTRs not annotated as "uncertain" in previously curated annotations[14] were included.

*K*-mer features were defined as the positions and counts of all *k*-mers in the length of either 5–6 nt or 4–7 nt within the specified regions of the 3′ UTR of each mRNA isoform for the linear regression model. For isoforms with 3′ UTRs shorter than the length of the specified region, the entire 3′ UTR was examined. Isoforms with 3′ UTRs shorter than 10 nt were excluded. Each 3′ UTR was appended with the tri-nucleotide

sequence "AAA" before $k$-mer extraction. This sequence was treated as part of the 3′ UTR for both $k$-mer feature calculation and the specified region length (e.g., for the "last 1000 nt" region, the last 997 nt of the annotated 3′ UTR were used before appending "AAA").

The number of each $k$-mer $N^k$ was counted with the *oligonucleotideFrequency* function from the R package *Biostrings*[61]. A composite positional feature value $P^k$ was computed for each $k$-mer to summarize its positional distribution relative to the 3′ end:

$$P^k = \begin{cases} \sum_j \frac{1}{\max(d_j, 1)}, & j \notin \varnothing \\ 0, & j \in \varnothing \end{cases} \quad (1)$$

where $d_j$ is the distance from the 3′ end to the $j$-th occurrence of the $k$-mer. The linear regression model predicting tail-length change $L_i$ for isoform $i$ was defined as:

$$L_i = c^0 + \sum_k c_N^k \cdot N_i^k + \sum_k c_P^k \cdot P_i^k \quad (2)$$

where $c^0$, $c_N^k$, and $c_P^k$ are the model coefficients to be optimized. All features were scaled using *RobustScaler* from the Python *scikit-learn* package[62]. Data were randomly stratified into 10 folds using *StratifiedKFold*[62], maintaining the proportion of isoforms with 0, 1, 2, 3, or ≥ 4 CPE (UUUUA) elements within their 3′ UTRs.

The model was implemented using a single Dense layer in *Tensorflow*[63] and trained with the Adam optimizer. Hyperparameters were tuned using the *Optuna*[64] framework. The negative $R2$ (between measured and predicted tail-length changes) for the validation data was used as the objective to minimize for optimization. Final hyperparameters were: learning_rate = 0.0025169, l1_regularizer = 0.000166, l2_regularizer = 0.0001097, epochs =100, batch_size = 50, loss = 'mse', patience = 10. Model performance was evaluated in 10-fold cross-validation. In this procedure, the 9-fold data were used for model training and validation (further split into 90% for training and 10% for validation, stratified by CPE count) while the remaining fold was used for testing the model performance. After each epoch, the model was evaluated on the validation set. Training stopped early if validation loss did not improve over 10 consecutive epochs, and the best model (based on the lowest validation loss) was saved. This training/validation procedure was repeated 5 times, accounting for differences in weight initializations, and the model with the highest $R^2$ (between measured and predicted tail-length changes) for the validation data was selected to predict the remaining fold. The procedure was rotated across all 10 folds. Final model performance was assessed by concatenating predictions from all test folds and comparing them with the measured tail-length changes.

## PAL-AI for poly(A) tail-length changes

The following datasets were used to train or validate different PAL-AI models: 1) the frog endogenous mRNA dataset: same as that used in the linear regression model; 2) the N60(LC)-PAS*mos* library dataset: tail-length changes of the N60(LC)-PAS*mos* library mRNAs injected into frog oocytes, comparing median tail lengths measured at 0 h and 7 h post-progesterone treatment during oocyte maturation, requiring at least 50 poly(A) tags at both time points; 3) the single-nucleotide mutagenesis library dataset: tail-length changes of the single-nucleotide mutagenesis library mRNAs injected into frog oocytes, comparing median tail lengths measured at 0 h and 7 h post-progesterone treatment during oocyte maturation, requiring at least 50 poly(A) tags at both time points; 4) the mouse endogenous mRNA dataset: tail-length changes of mouse endogenous mRNAs, comparing between median tail lengths measured in GV and MII oocytes, requiring at least 50 poly(A) tags in both replicate-merged datasets as described previously[14]; 5) the human endogenous mRNA dataset: tail-length

changes of human endogenous mRNAs, comparing between median tail lengths measured in GV and MII oocytes, requiring at least 50 poly(A) tags in both datasets as described previously[14].

To encode sequences as input for PAL-AI, RNA sequences from the specified regions of each mRNA isoform's 3′ UTR were appended at the 3′ end with the tri-nucleotide sequence "AAA". For isoforms with 3′ UTRs shorter than the specified region, the entire 3′ UTR was used, and "N"s were padded at the 5′ end. Isoforms whose AAA-appended 3′ UTRs were shorter than 10 nt were excluded. Sequences were one-hot encoded into four channels, with "N" represented by a zero vector. When coding regions were included, they were padded to the 5′ end of the AAA-appended 3′ UTRs. If the total length was still shorter than the required input length, "N"s were padded at the 5′ end. A fifth channel was added, where a value of 1 marked 3′-UTR positions and 0 marked coding or padded regions. When secondary structure information was included, *RNAfold*[32] was used to predict structure from the AAA-appended 3′ UTR. The unpairing probability of each nucleotide was added as a sixth input channel. Nucleotides from the coding region and padded "N"s were assigned a value of 0 in this sixth channel. The inputs for different final PAL-AI models were summarized in Supplementary Data 1.

The base PAL-AI model PAL-AI-s integrated a convolutional neural network with a recurrent neural network, in an architecture similar to Saluki[31]. It consisted of four modules: 1) Input block, which included an input layer, a 1D convolution, a normalization layer, and an activation layer. 2) Convolutional block, which included repeated structure, with each repeat comprising a 1D convolution layer, a normalization layer, an activation layer, a 1D maxpooling layer, and a dropout layer. 3) Recurrent block, which included a recurrent layer (GRU, BiGRU, LSTM, or BiLSTM), a normalization layer, and an activation layer. 4) Dense block, which contains a dense layer, a dropout layer, a normalization layer, an activation layer, and a final dense layer with a single output for the tail-length change. All convolution layers used the same number of filters and kernel size. The convolutional layers, the recurrent layer, and the first dense layer used the same kernel initializer and regularizer. All dropout layers had the same dropout rate. Layer normalization was used in all blocks, except for the Dense block, in which batch normalization was used. All activation layers used the same function—either SELU, SiLU (Swish), or Leaky ReLU. If SELU was used, the Lecun normal initializer and Alpha Dropout were applied. Otherwise, the HE normal initializer and regular Dropout were used.

The PAL-AI-s model was implemented using the Python *TensorFlow* package[63] and trained with the Adam optimizer. Loss functions used were either the mean squared error (MSE) or the mean absolute error (MAE). The hyperparameter tuning, cross-validation strategy, and model performance evaluation followed the same procedures as the linear regression model.

For the N60(LC)-PAS*mos* library, PAL-AI-s used the same architecture. Sequences in the 60-nt random region and constant region were appended with "AAA" and one-hot encoded. Training and evaluation were performed as described for endogenous mRNAs.

The PAL-AI-m and PAL-AI-ms models shared the same base architecture as PAL-AI-s, with three key modifications. First, an additional input feature was included to indicate the dataset group for each mRNA sequence. Second, rather than a single output, the final dense layer outputs multiple values, with one per dataset group. Third, a custom gathering layer was added after the dense layer to select the appropriate group-specific output based on the input dataset label. For the PAL-AI-m model, the frog endogenous mRNA dataset (output head 1) and the N60(LC)-PAS*mos* library dataset (output head 2) were used. For the PAL-AI-ms model, the frog endogenous mRNA dataset (output head 1), the human endogenous mRNA dataset (output head 2), and the mouse endogenous mRNA dataset (output head 3) were used. All input sequences were encoded as described for PAL-AI-s. Training and evaluation followed the same

procedure as PAL-AI-s, with the following differences to accommodate multiple groups. First, when splitting the training and validation set, the validation set contributed 10% of its data for the group with the smallest dataset, and validation sets from all other groups were downsampled to match this number. Second, for training, data from larger groups were similarly downsampled per epoch to match the smallest group, and batches were alternated across groups. Third, the average of negative $R2$ values (between measured and predicted tail-length changes) across validation groups was used as the objective to minimize during hyperparameter optimization.

A ResNet-based variant of PAL-AI was also tested, replacing the second module of PAL-AI-s with a dilated convolutional ResNet architecture[33] comprising seven groups with dilation rates of 1, 2, 4, 8, 4, 2, and 1. Each group contained four blocks, each consisting of a dilated convolution layer, layer normalization, an activation layer, a second dilated convolution layer, another layer normalization, a residual skip connection, and a final activation layer.

For all PAL-AI models, an optional input representing the initial tail length could be provided. When included, this value was concatenated to the input of the final dense layer. Although incorporating this feature improved performance for some datasets, it reduced generalizability, as initial tail-length measurements were not available for all mRNAs.

For all predictions made with PAL-AI, input sequences were encoded using the same scheme as during model training. To compare model performance on the single-nucleotide mutagenesis library, predictions were generated using either PAL-AI-s (endogenous mRNA-trained) or PAL-AI-m (endogenous mRNA-specific output head 1). Unless otherwise specified, final predictions were computed as the average output across the best-performing models from each fold of the 10-fold cross-validation to reduce fold-specific or initialization-specific biases and improve overall generalizability.

Hyperparameters for each PAL-AI model were optimized using *Optuna*[64], with final values summarized in Supplementary Data 1.

**In silico mutagenesis (ISM).** For frog and human endogenous mRNAs, all 3′ UTRs were analyzed except those previously annotated as "uncertain"[14]. For the N60(LC)-PAS$^{mos}$ library, all 3′ UTRs in the curated reference were used. Input sequences were constructed and encoded as described for PAL-AI. Tail-length changes of the wild-type sequences were predicted using PAL-AI. For ISM, each nucleotide within 3′ UTRs was individually substituted with each of the three possible alternative nucleotides (the fifth channel annotating the 3′ UTR was not altered), and the tail-length change of each mutant was predicted with corresponding PAL-AI models: PAL-AI-s (frog endogenous mRNA-trained) for frog endogenous mRNAs, PAL-AI-s (N60(LC)-PAS$^{mos}$ library-trained) for N60(LC)-PAS$^{mos}$ library mRNAs, and PAL-AI-ms (human-specific output head 2) for human endogenous mRNAs. The difference between the predicted tail-length change of the wild-type and each mutant was then calculated.

To examine the tail-length effect of losing or gaining a $k$-mer, differences between the predicted tail-length change of the wild-type and those of all single-nucleotide substitution mutants that disrupted the original $k$-mer or introduced a new $k$-mer were averaged across all instances of that $k$-mer within the last 300 nt of 3′ UTRs from all mRNA isoforms.

In some cases, $k$-mer-associated differences in predicted tail-length changes were examined in an iterative exclusion process. Only mutations within the last 300 nt of 3′ UTRs were considered. In each round of the iteration, mutations that disrupted (for $k$-mer loss) or introduced (for $k$-mer gain) any $k$-mer already present in an exclusion list (initially empty) were ignored. The average difference in predicted tail-length change between the wild-type and the mutant was then calculated for all $k$-mers that were not in the exclusion list. The $k$-mer

with the largest tail-length increase (for $k$-mer gain) or decrease (for $k$-mer loss) was recorded and added to the exclusion list before starting the next round. The iteration continued until no $k$-mer showed a statistically significant increase (or decrease) in predicted tail-length change (Welsh's t-test; Bonferroni-adjusted $P > 0.05$).

Motif logos were generated from 8-mers associated with poly(A) tail-length changes. $K$-mers were filtered by three metrics: the tail-length difference (positive for increase or negative for decrease), the Z-score of this difference ($\geq 3$ or $\leq -3$), and the Bonferroni-corrected $P$ value from Welch's t-test ($<0.01$). Tail-length changes were used as weights for generating motif logos. Retained $k$-mers were ranked by tail-length changes in descending order when examining motifs associated with tail-length increases, and in ascending order when examining motifs associated with tail-length decreases. An iterative clustering procedure was applied to group similar $k$-mers. Starting with the top-ranked $k$-mer, each subsequent $k$-mer was aligned to all entries in a growing seed list (initially empty). The first $k$-mer was added to the seed list without alignment. Alignments were ungapped and scored across all possible positions using the following scheme: +1 for a match, $-1.5$ for a mismatch, and $-1$ for a position offset. The alignment with the highest score (required to be $>-0.1$) was retained, and the $k$-mer's weight was assigned to this alignment. If multiple alignments tied for the highest score—whether due to different positions or different seed matches—all tied alignments were retained, and the weight was evenly distributed among them. If no alignment exceeded the threshold score, the $k$-mer was considered unaligned and added to the seed list as a new seed. This procedure continued until all retained $k$-mers had been either aligned to an existing seed or added to the seed list. For each seed and its aligned $k$-mers, a position weight matrix (PWM) was computed. At each alignment position, nucleotide-specific weights were summed across all aligned $k$-mers. If a $k$-mer did not contribute a nucleotide at a given position, its weight was distributed equally among A, C, G, and U. The summed weights were then normalized by the total weight at each position to yield nucleotide probabilities. Motif logos were generated from the resulting PWMs using the R package *ggseqlogo*[65].

## Motif insertion analysis

Filtering and encoding of input sequences were performed as in the ISM analysis. To insert a motif, the sequence of the same length at a specified position within the 3′ UTR of an mRNA was replaced with the motif. The fifth channel annotating the 3′ UTR was left unchanged. PAL-AI was then used to predict the tail-length changes of both the wild-type and motif-inserted mutant. The difference between these predictions was used as the insertion outcome. When analyzing the relative positioning of the PAS and CPE elements, only mRNAs whose 3′ UTRs contained at least a PAS within the last 100 nt were included. For these mRNAs, the relative distance was calculated between each CPE and the PAS closest to the 3′ end.

## Reanalysis of reporter libraries for $k$-mer-associated tail-length changes

Poly(A) tail-length measurements of the frog oocyte-injected N60-PAS$^{mos}$ library at 0 h and 7 h post-progesterone treatment and CPE$^{mos}$-N60 library at 0 h and 5 h post-progesterone treatment were obtained from a previous study[14]. For the N60-PAS$^{mos}$ library, the analysis was restricted to reporter variants containing exactly one instance of the UUUUA motif within the 3′ UTR. For the CPE$^{mos}$-N60 library, the analysis was restricted to reporter variants not containing any AAUAAA or AUUAAA motifs within the 3′ UTR. For each $k$-mer, all variants containing that $k$-mer were identified, and the mean tail length across those variants was computed. To assess the positional effect of specific $k$-mers, only variants in which the $k$-mer appeared at a defined position were included in the calculation.

## Preparation of the N60(LC)-PAS$^{mos}$ library

The DNA template for the N60(LC)-PAS$^{mos}$ library was generated by PCR using primers KXU024 and KXU068, and 2 amol of the N60-PAS$^{mos}$ DNA template used previously[14]. The reaction (100-µl total volume) was performed with the KAPA HiFi HotStart Kit (Roche, KK2502). PCR products were gel-purified using agarose gels (Lonza, 50004) and the GeneJet Gel Extraction Kit (Thermo Fisher, K0692).

For mRNA library preparation, in vitro transcription was carried out in a 100-µl reaction containing 40 mM Tris (pH 8.0), 21 mM MgCl$_2$, 2 mM spermidine (Sigma, 85558-1 G), 1 mM dithiothreitol (GoldBio, DTT25), 5 mM NTP Mix (Thermo Fisher, R0481), 0.2 U yeast inorganic pyrophosphatase (New England Biolabs, M2403L), 80 U SUPERase·In (Thermo Fisher, AM2694), 2 µg DNA template, and T7 RNA polymerase (purified in-house, final concentration 6.4 ng/µl). The reaction was incubated at 37 °C for 3 h. To remove DNA templates, 2 U of DNase I (New England Biolabs, M0303S) were added, followed by a 20-minute incubation at 37 °C. To enhance HDV ribozyme cleavage, thermal cycling was performed (65 °C for 90 s, followed by 37 °C for 5 min, repeated for four cycles in 50-µl aliquots per tube). Before gel loading, 2 µl of 0.5 M EDTA (pH 8.0) and 100 µl 2× Gel Loading Buffer II (Thermo Fisher, AM8547) were added. Samples were incubated at 65 °C for 5 min, then resolved on 5% urea–acrylamide denaturing gels. Desired RNA bands were visualized by UV shadowing, excised, macerated, and eluted overnight (>16 h) at 23 °C in 10 mM HEPES (pH 7.5) and 300 mM NaCl with shaking (1400 rpm, 15 s on/105 s off) on a thermomixer. Gel debris was removed using Spin-X columns (Corning, 8160), and RNA was precipitated with isopropanol and resuspended in water.

RNA capping was performed using the Vaccinia Capping System (New England Biolabs, M2080S) according to the manufacturer's protocol, with the capping enzyme used at 2 U/µl. Capped RNAs were purified by phenol/chloroform extraction and ethanol precipitation, then desalted using Micro Bio-Spin P-30 columns (Bio-Rad, 7326250). To remove 2′,3′-cyclic phosphates generated by HDV ribozyme cleavage, capped RNAs (up to 100 µg) were incubated in a 100-µl reaction containing 50 U T4 polynucleotide kinase (PNK; New England Biolabs, M0201S), 1× T4 PNK buffer, and 25 U SUPERase·In at 37 °C for 1 h. RNAs were then purified again by phenol/chloroform extraction and ethanol precipitation.

Finally, RNAs were resuspended in 1× Gel Loading Buffer II and further purified using urea–acrylamide denaturing gels as described above. RNA integrity was verified by visualization on formaldehyde–agarose denaturing gels as described[11], and samples were stored at −80 °C until use.

## Preparation of the single-nucleotide mutagenesis library

The oligo pool was synthesized as SurePrint HiFi Oligo (Agilent Technologies; Supplementary Data 3). Each oligo contained a variable region derived from the last 100 nt of the 3′ UTR of one of ten frog oocyte mRNAs (*atp1a1.S*, *ccnb1.2.L*, *ccnb2.L*, *lima1.L*, *mad2l1.L*, *magoh.S*, *mos.L*, *tpx2.L*, *aurkaip1.L*, and *dbf4.L*). At each position, every nucleotide was mutated to the three alternative nucleotides, yielding 3010 sequences (10 wild-type and 3000 mutants). A 35-nt poly(A) fragment was appended to the 3′ end, and constant sequences (24 nt at the 5′ end and 20 nt at the 3′ end) were added for PCR amplification. Both forward and reverse strands were included in the pool.

The DNA template for in vitro transcription was assembled by overlapping PCR using the KAPA HiFi HotStart Kit. Three fragments were joined sequentially. Fragment 1 (F1) was amplified from the N60-PAS$^{mos}$ template[14] with primers KXU024 and KXU236. Fragment 2 (F2) was amplified from the oligo pool with primers KXS067 and KXU237. Fragment 3 (F3) was amplified from the plasmid C071[11] with primers KXU110 and KXU068. F1 and F2 were first joined, and the resulting product was joined with F3 in a 100-µl PCR reaction containing 10 pmol of each fragment, 2 µl KAPA HiFi enzyme, 3 µl 10 mM dNTP, and 20 µl 5× KAPA HiFi buffer for 15 cycles, with an annealing temperature of 66 °C.

PCR products were purified using agarose gels and the GeneJet Gel Extraction Kit. The mRNA library was then in vitro transcribed and capped, followed by removal of the 3′-end cyclic phosphate as described for the N60(LC)-PAS$^{mos}$ library.

## Preparation of the reporter mRNA poly(A) tail-length standard mix

Each of the four poly(A) tail-length standards was constructed from two fragments (F1 and F2) joined by overlapping PCR. Both fragments were amplified from plasmid C071 using the KAPA HiFi HostStart Kit and the following primer sets: KXU024 and one of KXU290, KXU291, KXU292, or KXU293 (for F1); KXU068 and one of KXUm019, KXUm020, KXUm021, or KXUm022 (for F2, yielding tail lengths of 10, 50, 90, or 130 nt). Overlapping PCR was performed in a 50-µl reaction containing 100 fmol of each fragment, 1 µl KAPA HiFi enzyme, 1.5 µl 10 mM dNTPs, and 10 µl 5× KAPA HiFi buffer. After 5 cycles (annealing temperature 63 °C), primers KXU024 and KXU068 were added to a final concentration of 300 nM along with 1 µl KAPA HiFi enzyme, 1.5 µl 10 mM dNTP, and 10 µl 5× KAPA HiFi buffer to bring the final volume to 100 µl. Twenty-four additional PCR cycles were performed. PCR products were purified using agarose gels and the GeneJet Gel Extraction Kit. Each RNA was in vitro transcribed and capped, followed by removal of the 3′-end cyclic phosphate as described for the N60(LC)-PAS$^{mos}$ library. The four mRNA standards were mixed at equimolar concentrations. Each contained a unique 4-nt barcode in the 3′ UTR, introduced by KXU290, KXU291, KXU292, or KXU293. Sequences of the four tail-length standards are provided in Supplementary Data 4.

## Reporter mRNA library injection and sample collection

Frog oocytes were obtained from Xenopus1 (12005). Healthy Stage V–VI oocytes were hand-picked and transferred to OR-2 buffer (5 mM HEPES pH 7.6, 82.5 mM NaCl, 2.5 mM KCl, 1 mM MgCl$_2$, 1 mM CaCl$_2$, 1 mM Na$_2$HPO$_4$) supplemented with 100 µg/ml Gentamicin (Thermo Fisher, 15750060) and incubated at 18 °C overnight (>16 h) for recovery. Injections were performed at 23 °C using a PLI-100 Plus Pico-Injector. For both the N60(LC)-PAS$^{mos}$ library and the single-nucleotide mutagenesis library, 4 nl of mRNA (0.1 pmol/µl) was injected per oocyte.

Frog oocytes were matured in vitro in OR-2 buffer supplemented with 10 µg/ml progesterone (Millipore Sigma, P0130), diluted from a 10 mg/ml ethanol-dissolved stock. Maturation timing varied across batches: in most cases, 50% of oocytes underwent germinal vesicle breakdown (GVBD; indicated by a white spot on the animal pole) within 3–5 h, and 100% reached GVBD by 7 h post-progesterone addition.

At indicated time points post-progesterone treatment, groups of 100 frog oocytes were collected. After removing OR-2 buffer, oocytes were washed three times with ice-cold buffer RL (20 mM HEPES pH 7.5, 100 mM KCl, 5 mM MgCl$_2$, 1% Triton X-100, 100 µg/ml cycloheximide, cOmplete protease inhibitor cocktail [MilliporeSigma, 11836170001, 1 tablet/10 ml], 10 µl/oocyte). Following the washes, buffer was completely removed, and oocytes were lysed in buffer RL (10 µl per oocyte) supplemented with 200 U/ml SUPERase•In (10 µl/oocyte) by vigorous shaking and pipetting. Lysates were clarified by centrifugation at 5000 × $g$ for 10 min at 4 °C. Supernatants were mixed with 3 volumes of TRIzol LS (Thermo Fisher, 10296010).

The 0 h post-progesterone sample was prepared by collecting the supernatant from lysed, untreated oocytes and mixing it with the uninjected mRNA library prior to TRIzol LS addition. All TRIzol-LS RNA extractions were performed using Phasemaker tubes (Thermo Fisher, A33248) following the manufacturer's instructions. Total RNA was resuspended in 40 µl water (0.4 µl/oocyte).

## Library preparation and sequencing of the injected mRNA libraries

Total RNA (10 µl, ~30 µg) from injected oocytes was combined with 10 amol of reporter mRNA poly(A)-tail-length standard mix. To this, 8

pmol each of oligos KXSH009 and KXSH010, and 2× SSC (0.3 M NaCl, 30 mM sodium citrate pH 7.0), were added in a final volume of 50 μl. The RNA-oligo mixture was incubated at 70 °C for 5 min and then cooled to 23 °C at a rate of 0.1 °C/sec for annealing. The annealed RNA was incubated with 40 μl MyOne Streptavidin C1 beads (Thermo Fisher, 65002) for 20 min at 23 °C on a thermal mixer (15 s on and 1 min 45 s off). The supernatant was separated from the beads with a magnetic rack and removed. Beads were washed twice with 300 μl 1× B&W buffer (5 mM Tris-HCl pH 7.5, 0.5 mM EDTA, 1 M NaCl) and once with 300 μl 2× SSC. RNA was eluted from beads with two sequential incubations: first in 100 μl 10 mM HEPES pH 7.5 at 65 °C for 3 min, then in 100 μl water at 65 °C for 3 min. Eluates were combined, ethanol-precipitated, and resuspended in 6.5 μl water.

Anti-sense oligo-enriched RNA was ligated to a pre-adenylated 3′ adapter (KXS347a) in a 10-μl reaction containing 5 μM adapter, 50 mM HEPES pH 7.5, 10 mM MgCl₂, 10 mM dithiothreitol, and 1 U/μl T4 RNA ligase 1 (NEB, M0204S). Ligation was performed at 23 °C for 150 min. Ligated RNA was extracted with phenol/chloroform, ethanol-precipitated, and resuspended in 11.4 μl water. The ligated RNA was mixed with 0.6 μl 100 μM primer KXS348 and 1 μl 10 mM dNTPs in a total volume of 13 μl, incubated at 65 °C for 5 min, and cooled on ice for 1 min. Reverse transcription was performed in a 20-μl reaction using SuperScript IV (ThermoFisher, 18090050), with 1× SSIV Buffer, 5 mM dithiothreitol, 1 U/μl SUPERase•In, and 200 U enzyme, incubated at 55 °C for 15 min. After reverse transcription, RNA was then hydrolyzed with 3.3 μl 1 M NaOH at 90 °C for 10 min, followed by neutralization with 24 μl 1 M HEPES pH 7.5. The resulting cDNA was ethanol-precipitated, resuspended in water, and amplified in a 50 μl PCR reaction using primer KXS037 and a barcoded primer (KXS057) targeting a constant region of the library. Amplification was performed using the KAPA HiFi HotStart Kit following the manufacturer's suggested protocol for 11-14 cycles. The final PCR-amplified product was cleaned up twice with AMPure XP beads (Beckman Coulter, A63881) at a 1.2× beads-to-sample ratio.

Sequencing was performed on an Illumina HiSeq 2500, with a custom run of two reads: a 281-cycle first read using primer KXS067, and a 10-cycle second read using a standard TruSeq sequencing primer. To generate a reference, the uninjected 0 h N60(LC)-PAS$^{mos}$ library sample was also sequenced on AVITI using the same primers, with custom read lengths of 299 and 10 cycles.

**Variant and tail-length analysis for the injected mRNA libraries**

To build a reference sequence for the N60(LC)-PAS$^{mos}$ library, read 1 from the 0 h post-progesterone (uninjected) sample sequenced on AVITI was trimmed to 81 nt from the 5′ end. Trimmed reads were screened for the presence of a known constant region from the 3′ UTR of poly(A) tail-length standards (ACCAGCCTCAAGAACACCCGA ATGG). A maximum of seven mismatches was allowed within the first 25 nt of this sequence, permitting positional offsets from −3 to +3 nt. Next, the remaining reads were examined for the expected constant sequence AATAAAGAAATTGATTTGTCT at positions 61–81, again allowing offsets from −3 and +3 nt. Reads containing a 21-nt segment with no more than six mismatches to the constant sequence at any allowed position were retained. For reads with a non-zero offset, alignment was corrected to zero: if the offset was negative, "N" bases were prepended to the read; if positive, the corresponding number of bases was trimmed from the start of the read. These modified reads were then clustered using *UMICollapse*[66], and clusters with more than 100 supporting reads were retained. The consensus (primary) sequence from each cluster was used to construct the reference sequence for the N60(LC)-PAS$^{mos}$ library.

During the HiSeq 2500 run for both the N60(LC)-PAS$^{mos}$ and single-nucleotide mutagenesis libraries, a sequencing artifact occurred at cycle 26 of read 1, resulting in a repeated base call at cycle 26 and 27. Consequently, base 27 was removed from all reads prior to downstream analysis.

For the N60(LC)-PAS$^{mos}$ libraries, read 1 sequences from the HiSeq run were aligned to the reference using *STAR* (v2.7.1)[67] with the following parameters '--outFilterMultimapNmax 1 --alignEndsType Local −clip3pNbases 199 --outSAMattributes All --outSAMtype BAM Sorted-ByCoordinate'. Poly(A)-tail lengths for mapped reads were determined as previously described for the N60-PAS$^{mos}$ libraries[14].

For the single-nucleotide mutagenesis libraries, read 1 sequences from the HiSeq run were aligned to the reference consisting of the last 100 nt of the 3′ UTR from ten selected frog oocyte mRNAs (wild-type sequences) using *STAR* (v2.7.1)[67] with these parameters '--out-FilterMultimapNmax 1 --outFilterMismatchNmax 1 --alignEndsType EndToEnd −clip3pNbases 180 --outSAMattributes All --outSAMtype BAM SortedByCoordinate'. Mutations were classified using the CIGAR string in the resulting BAM file. Poly(A)-tail lengths for these mapped reads were determined as for the N60(LC)-PAS$^{mos}$ libraries, with the tail-start position defined as position 101 of read 1. Only wild-type and single-nucleotide substitution variants were retained. For each variant, reads sharing the same read 2 sequences (serving as a unique molecular identifier, UMI) were grouped and collapsed. The median tail length of each collapsed was used as the representative poly(A)-tail length.

Reference sequences of both the N60(LC)-PAS$^{mos}$ and single-nucleotide mutagenesis libraries are provided in Supplementary Data 4.

**RNA secondary structure prediction**

RNA secondary structures for the 3′ UTR sequences in the single-nucleotide mutagenesis library were predicted using *EternaFold* (v1.3.1)[43] with the parameters "--params parameters/EternaFoldParams.v1". The resulting dot-bracket notations were visualized using *RNAcanvas*[68].

**Analysis of mouse oocyte data sequenced by Nanopore**

Poly(A)-tail lengths of mouse oocyte mRNAs obtained via Oxford Nanopore Technologies (ONT) sequencing were retrieved from a previous study[13]. Tail-length changes were calculated by comparing the median tail lengths between GV and MII oocytes, requiring at least 50 poly(A) reads in datasets from both stages. As ONT-reported tail-length measurements were reported per gene rather than per mRNA isoform, comparisons were made with the primary isoform for each gene, defined based on HiSeq data[14] or predicted using PAL-AI-ms (mouse-specific output head 3). Analyses were restricted to genes whose primary isoform represented more than 90% of expression in GV oocytes.

**Prediction of poly(A) tail-length changes of reporter mRNAs during mouse oocyte maturation**

To evaluate the relationship between predicted tail-length changes and experimentally measured translational efficiency changes of reporter mRNAs injected into mouse oocytes, source data were obtained from a previous study (see Fig. 4g of the original publication)[12]. Translational efficiency change was calculated as the difference between the mean normalized fluorescence signals in GV versus MII oocytes (24 h). Reporter mRNA 3′ UTR sequences were inferred from the cloning and mutagenesis primers used in this study. Tail-length changes for these sequences were predicted using PAL-AI-ms (mouse-specific output head 3).

**Cross-species analysis of poly(A) tail-length and translational efficiency changes**

Gene orthology tables (frog-to-human from Xenbase, mouse-to-human from Ensembl) were used to map homologous genes. Measured tail-length changes were compared between GV and MII oocytes in humans and mice, and between 0 h and 7 h post-progesterone treatment in frog oocytes. For mRNA isoforms with unique 3′ ends in one of the three species, predicted tail-length changes from PAL-AI-ms were used if experimental data were unavailable. For genes with multiple mRNA

isoforms, the dominant isoform (>50% by tag count in GV oocytes of humans or mice, or in 0 h post-progesterone frog oocytes) was selected. If a human gene had multiple homologs in frog or mouse, the average tail-length change across all the homologs was used.

Translational efficiency data derived from ribosome profiling and mRNA-seq were reprocessed[14] and compared between GV and MII oocytes in humans[6] and mice[12].

For gene ontology (GO) analysis, genes showing substantial tail lengthening (≥15 nt, by measurement or prediction) during oocyte maturation in all three species were selected. These genes were ranked by the smallest tail-length change across three species, and the ranked list was submitted to the *gost* function in the R package *gprofiler2*[69] with the parameters "ordered_query = TRUE, user_threshold = 0.05, correction_method = "g_SCS". The top eight non-overlapping GO:Biological Process categories (not from the same parent term and ranked by gSCS-adjusted *P* value) were reported.

### Evolutionary constraint associated with tail-length changes

PhyloP scores from multiple sequence alignments of 99 vertebrate genomes to the human genome were obtained from the UCSC Genome Browser[46]. For each nucleotide within the last 100 nt of human 3′ UTRs with an available phyloP score, the mean PAL-AI-predicted differences in tail-length changes when this nucleotide was substituted with each of the three alternative nucleotides (from ISM analysis) were calculated.

### Prediction of poly(A) tail-length changes for human genetic variants from the All of Us Research Program and gnomAD

MRNA 3′ UTR isoforms shorter than 10 nt were excluded from analysis. To maximize variant coverage despite incomplete transcript annotations, the "uncertain" 3′ UTRs in the annotations were retained if their 3′ ends were supported by entries in the PolyA_DB_v3.2 database[70].

For the data reported in the All of Us Research Program[47], all single-nucleotide polymorphism and indel variants from short-read whole-genome sequencing callsets encompassing human exome regions (version 8) were intersected with human oocyte 3′ UTR genomic coordinates. Variants overlapping these regions were selected. For each variant, the associated 3′ UTR sequence was reconstructed by substituting the reference allele with the alternative allele, followed by appending three adenines (AAA) at the 3′ end. Tail-length changes were predicted using PAL-AI-ms (human-specific output head 2). The mutational outcome was defined as the difference in predicted tail-length change between the variant and corresponding wild-type (reference) sequence. In cases where a variant overlapped multiple 3′ UTRs, the one with the largest absolute difference in predicted tail-length change was used for downstream analysis.

For the data reported in gnomAD[49], genomic variants from v4.1 (covering 76,215 individuals) were downloaded from gnomad.broadinstitute.org. Variants were included if they passed all quality filters (indicated by "PASS" in the "FILTER" column in the VCF file) and intersected annotated 3′ UTR genomic locations. Tail-length changes and mutational outcomes were computed using the same approach as for the All of Us Research Program.

### Quantification and statistical analysis

Graphs were generated and statistical analyses were performed using R[71]. Statistical parameters, including the value of n, statistical test, and statistical significance (*P* value), are reported in the figures or their legends, or Supplementary Data 5. No statistical methods were used to pre-determine sample size.

### Reporting summary

Further information on research design is available in the Nature Portfolio Reporting Summary linked to this article.

## Data availability

All standard sequencing data generated in this study are available in the Gene Expression Omnibus under the accession number GSE280422. Raw intensity data for reporter mRNA tail-length sequencing cannot be deposited in public databases due to their large sizes and are available upon request. Other processed data are available at Zenodo with https://doi.org/10.5281/zenodo.15461000, except for that derived from gnomAD and the All of Us Research Program, due to privacy policies on human genetic information. Oligo sequences used in this study are listed in Supplementary Data 2. Sequences of the oligo library used for the single-nucleotide mutagenesis library are listed in Supplementary Data 3. Sequences of the N60(LC)-PAS^mos library, the single-nucleotide mutagenesis library, and the tail-length standards are listed in Supplementary Data 4. Other publicly available data analyzed in this study are indicated in the relevant sections of Methods. To access the genomic data reported in the All of Us Research Program, a Controlled Tier account is required on the Workbench (https://workbench.researchallofus.org). Accession codes with links are listed below: GSE280422 [https://www.ncbi.nlm.nih.gov/geo/query/acc.cgi?acc=GSE280422] (N60(LC)-PAS^mos library mRNA and single-nucleotide mutagenesis library mRNA tail-length data), 15461000 [https://doi.org/10.5281/zenodo.15461000] (Processed data in this study), GSE228001 [https://www.ncbi.nlm.nih.gov/geo/query/acc.cgi?acc=GSE228001] (Mouse oocyte mRNA tail-length data measured by ONT), GSE241107 [https://www.ncbi.nlm.nih.gov/geo/query/acc.cgi?acc=gse241107] (frog oocyte mRNA, N60-PAS^mos library mRNA, and CPE^mos-N60 library mRNA tail-length data; reprocessed mouse and human oocyte translational efficiency data), HRA001911 [https://ngdc.cncb.ac.cn/gsa-human/browse/HRA001911] (human oocyte tail-length data), GSE197265 [https://www.ncbi.nlm.nih.gov/geo/query/acc.cgi?acc=GSE197265] (human oocyte ribosome-footprinting profiling and mRNA-seq data), GSE165782 [https://www.ncbi.nlm.nih.gov/geo/query/acc.cgi?acc=GSE165782] (mouse oocyte ribosome-footprinting profiling and mRNA-seq data), Figure 4g [https://static-content.springer.com/esm/art%3A10.1038%2Fs41556-022-00928-6/MediaObjects/41556_2022_928_MOESM4_ESM.xlsx] (source data for GFP signals of the GFP-*Snd1* 3′ UTR reporter presented in Fig. 6c), gnomAD v4.1 [https://gnomad.broadinstitute.org/data] (human genomic variant data reported in gnomAD), All of Us Research CDRv8 [https://workbench.researchallofus.org/] (human genomic variant data reported in All of Us Research Curated Data Repository Exome v8).

## Code availability

The code for PAL-AI is written in Python 3.8 and available at https://github.com/coffeebond/PAL-AI. Reporter mRNA tail-length sequencing data analyses were performed using a custom script written in Python 2.7 and available at https://github.com/coffeebond/MPRA_tail_seq. The codes for analyses and generating the figures are available at https://github.com/coffeebond/PAL-AI_paper.

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

## Acknowledgements

We thank Thy Pham, Peter Wang, and other current and former members of the Bartel lab for helpful discussions, Sumeet Gupta for assistant with data collection, and the Whitehead Institute Genome Technology Core for sequencing. D.P.B. is an investigator of the Howard Hughes Medical Institute.

## Author contributions

K.X. and D.P.B. designed the study and wrote the manuscript. K.X. conceived the project and performed all the experiments and analyses.

## Competing interests

The authors declare no competing interests.
