## [Transparent Peer Review file · Nature Communications]

PAL-AI reveals genetic determinants that control poly(A)-tail length during oocyte maturation, with relevance to human fertility

Corresponding Author: Dr David Bartel

Version 0:

Reviewer comments:

Reviewer #1

(Remarks to the Author)

In this manuscript, Xiang and Bartel present an intriguing study employing a novel machine-learning model named PAL-AI to predict poly(A) tail-length changes in maturing oocytes of frogs and mammals. The authors effectively demonstrate the model's capacity to accurately predict tail-length changes based on 3' UTR sequence information and its ability to identify critical regulatory elements and contextual features governing cytoplasmic polyadenylation during oocyte maturation. Additionally, the study introduces a clever approach for evaluating cis-regulatory elements in 3' UTRs through in-silico mutagenesis, which is validated by comparisons to both MPRA and endogenous experimental data.

The manuscript is well-written, and the study design is thorough and robust. The authors carefully address potential confounding factors by comparing stage-matched oocytes and implementing rigorous experimental protocols to minimize variability. However, the manuscript could better emphasize the novelty and impact of the proposed approach by addressing the following points:

1. Figures 3f, 4a, and 4b reveal that a few "C" positions contribute to specific deadenylation events, as predicted by PAL-AI and measured in MPRA. However, the interpretation of these findings is minimal and somewhat indirect. The manuscript would benefit from further clarification of whether these positions are observed in other reporters. Moreover, it would be helpful to include data on the abundance changes of reporters with single mutations at these sites to demonstrate the versatility of PAL-AI.
2. The dataset used for training the linear regression model, and PAL-AI is insufficiently described in the figures and text. Readers would find the study easier to follow if Figure 1 included a schematic diagram illustrating the experimental design and the training process of the machine-learning models.
3. Figure 6b shows no compelling reason to restrict the analysis to variants from the All of Us exome or gnomAD databases. Expanding the analysis to all transcript positions expressed in the human germline would reduce unnecessary complexity and improve the interpretability of the results.
4. In Figure 6c, the observed effect size seems too small, given the potential confounding factors related to the innate sequence bias of PAL-AI predictions. A more effective approach might involve comparing the ratio of singletons to more prevalent variants across several intervals of PAL-AI-predicted tail length changes. This would provide a clearer and more direct representation of the effect size.

(Remarks on code availability)

Reviewer #2

(Remarks to the Author)

In this work, Xiang and Bartel presented a deep learning model named PAL-AI to predict poly(A) tail length changes in

maturing oocytes. They showed that the PAS and CPE elements are major determinants of poly(A) tail lengthening. They performed the reporter assay by injecting mutated mRNAs into oocytes and showed that PAL-AI prediction is correlated with measured changes in mRNAs. The work is novel and valuable as there is no other deep-learning model developed to model cytosolic polyadenylation. I have some comments to improve the work.

1. For the model training, the authors used 1053 nt of 3'UTR sequences. The number looks arbitrary. The authors may try some other longer lengths such as 1200 nt, 1500 nt, and 2000 nt to show the robustness of algorithm performance. And for the deep learning model parameters in Fig. S1e, why do the authors use these specific filter sizes, kernel sizes, GRU units, and Dense units? The authors may try a few other numbers to show their selected parameters are optimal.
2. The authors showed that the canonical PAS AAUAAA promotes poly(A) tail lengthening. What about other PAS variants, such as AUUAAA, AGUAAA, and others? The authors can mutate AAUAAA to other variants in silico to examine the changes in the predicted values and use the reporter assay data to provide quantitative numbers comparing different PAS signals.
3. For the reporter assays, the authors showed that the predicted values are systematically shorter than measured tail-length changes. The authors discussed possible reasons. For the reporters, the starting RNAs all contain 35 As, which is different from endogenous genes with different lengths of starting As. Maybe the authors can add this possibility.
4. The authors discussed that some genes show differences comparing the predicted vs. measured values. One possibility is that the training sequences are biased and do not cover enough cases for these discrepancy cases to train a robust model. The authors can pool together the endogenous genes and reporter RNAs, retrain a model, and use the new model to evaluate the performance. This may help the author obtain a more robust model to cover some rare sequence cases explaining cytosolic polyadenylation.
5. For the genetic analysis, the authors showed selection pressure to avoid genetic variants altering the key motifs mediating cytosolic polyadenylation, such as AAUAAA and UUUUA. But these two motifs are also crucial for nuclear polyadenylation. UUUUA can be bound by Fip1 in the nucleus. The authors need to discuss this. The genetic constraints are not necessarily due to cytosolic polyadenylation.

(Remarks on code availability)

Well written with detailed instructions to run the codes.

Reviewer #3

(Remarks to the Author)

The manuscript introduces PAL-AI, a neural network designed to predict changes in poly(A) tail length in maturing frogs, mice, and human oocytes. By considering multiple sequences and contextual features influencing poly(A) tail length, this study offers a potentially robust approach to addressing a critical aspect of gene regulation during oogenesis.

The Authors provide new insights into the role of UGU/GUU-containing motifs in poly(A)-tail lengthening, which has previously been linked to the regulation of translation but not to the tail-length control. The analysis of genetic variants in human populations has potentially important implications for identifying genetic causes of female infertility. This work is timely and well-designed and has the potential to significantly advance our understanding of oocyte maturation and its impact on human fertility.

A notable strength of this study is the initial use of a simple linear regression model, while transitioning to a more sophisticated neural network only when justified by performance shortcomings. This deliberate choice underscores that the adoption of a neural network is a thoughtful decision, not driven by hype for machine learning. The Authors' clear rationale for the variable performance of the model enhances its credibility. While PAL-AI performs well for most mRNAs, it underperforms for some, particularly those with structural features affecting CPE or PAS accessibility. The authors acknowledge this limitation and propose a mitigation strategy in future models. The comprehensive description of the architecture ensures transparency. Furthermore, the authors effectively integrate previously acquired data, construct a robust reporter system, and utilize publicly available datasets, underscoring the versatility of their approach. This thorough validation demonstrates the potential for broad applicability of PAL-AI, given its ability to handle data from different organisms.

However, despite our overall positive impression, the work has several weaknesses that should be addressed before acceptance for publication.

We have one major concern:

The supplementary files available to reviewers needed the source data used to create the figures included in the text (e.g., data presented on scatterplots, results of go-term analysis, coordinates of individual motifs) and the complete input datasets passed to the neural network. With this, the reviewer cannot validate the biological data. This is the most serious deficiency that should be corrected if the work is to be published.

We have also some minor comments:

1. The manuscript lacks information on the validation of PAL-AI training, both in the supplementary materials and the main text. Including a confusion matrix, as well as loss and accuracy curves, would provide a more accurate assessment of the model quality and help identify potential overfitting.
2. Lines 841-845: It needs to be clarified why the Authors, after training and validating many models, did not choose the most optimal one but decided to average the results from multiple models. The text lacked a convincing explanation for this strategy.
3. To date, numerous datasets examining the dynamics of poly(A) tails in the germline using long-read sequencing have been published. Comparing the results obtained in this paper with available data from PacBio or Oxford Nanopore would provide orthogonal validation for tail length measurements. Specifically, comparing results with direct RNA sequencing data, which are free of amplification bias, would be a particularly valuable validation of the proposed model.
4. This study analyses data from evolutionarily distant species (frog, mouse, human) using a neural network model trained exclusively on frog data. If the goal was to create a universal model, the training set design should have been more carefully considered. Restricting training to frog data introduces errors in mammalian-specific nucleotide contexts. Alternatively, organism-specific models or at least amphibian and mammalian variants should be developed. Scatterplots indicate the amphibian-trained model underperforms in mammalian cells (Fig. 1a vs. 5a, b). Although PAL-AI shows slightly higher correlations than the initial regression-based approach, this may stem from overplotting (tight data grouping) rather than a significant trend. Using species-specific or taxon-specific models would enable a more detailed investigation of potential patterns in the analyzed data.
5. Was the data used to train the model curated? Specifically, were estimates of the relative positions of potential motifs compared to data available in databases/repositories? How reliable are the 3'UTR sequences that were used as references?

(Remarks on code availability)

1. The software is available at the provided link and can be installed without issues. It includes a test dataset that enables users to verify its functionality. However, when analyzing the test data, the `INN_main.py` script generates a warning at line 733. While this warning does not directly affect the predictions, it suggests that the check for 'NA' values may not be functioning as intended. Addressing this issue will ensure the script handles 'NA' values in the `Y_df[y]` column correctly.
2. Adding a repository section explaining each column in the final output (classification results) or using more intuitive table headers would help audiences less familiar with machine learning outputs. This would also simplify downstream processing for general users.
3. The use of the SELU function (instead of the commonly used RELU) is clever, allowing it to counteract vanishing and exploding gradient problems and, thus, a decrease in the quality of model predictions. However, has weight initialization been implemented properly? The SELU function requires a specific kernel initializer, as pointed out in the TensorFlow docs. Has the dropout type also been adjusted? The code available in the GitHub repository uses the default initializer and dropout, which is not a very fortunate choice if one does not want SELU to lose its self-normalizing properties.
4. The pipeline supports an alternative network architecture, ResNet, but this is not mentioned in the manuscript. The absence of a comparison between the described architecture and ResNet results is a notable omission and leaves the reviewer somewhat dissatisfied.

Reviewer #4

(Remarks to the Author)

(Remarks on code availability)

Version 1:

Reviewer comments:

Reviewer #1

(Remarks to the Author)

I have carefully reviewed the revised manuscript and appreciate the authors' efforts to address the feedback provided. The responses to my questions and requests have been satisfactory, and the concerns raised by other reviewers appear to have been adequately addressed. The addition of Figure 7a,c-d is particularly noteworthy, as it provides valuable evidence for the validity of PAL-AI model from a different perspective, strengthening the overall contribution of the work.

I believe the manuscript has reached a suitable standard for publication.

(Remarks on code availability)

Reviewer #2

(Remarks to the Author)

The authors did a great job addressing my comments. I am happy to recommend this paper to be published in the current form.

(Remarks on code availability)

Reviewer #3

(Remarks to the Author)

The Authors adequately addressed all the issues raised by the reviewers. Therefore I recommend this manuscript for publication.

(Remarks on code availability)

Reviewer #4

(Remarks to the Author)

(Remarks on code availability)

All of my remarks were addressed carefully. The authors made an excellent paper out of the very good paper. I recommend this work for publication.

Reviewer #1 (Remarks to the Author):

In this manuscript, Xiang and Bartel present an intriguing study employing a novel machine-learning model named PAL-AI to predict poly(A) tail-length changes in maturing oocytes of frogs and mammals. The authors effectively demonstrate the model's capacity to accurately predict tail-length changes based on 3' UTR sequence information and its ability to identify critical regulatory elements and contextual features governing cytoplasmic polyadenylation during oocyte maturation. Additionally, the study introduces a clever approach for evaluating cis-regulatory elements in 3' UTRs through in-silico mutagenesis, which is validated by comparisons to both MPRA and endogenous experimental data.

The manuscript is well-written, and the study design is thorough and robust. The authors carefully address potential confounding factors by comparing stage-matched oocytes and implementing rigorous experimental protocols to minimize variability. However, the manuscript could better emphasize the novelty and impact of the proposed approach by addressing the following points:

1. Figures 3f, 4a, and 4b reveal that a few "C" positions contribute to specific deadenylation events, as predicted by PAL-AI and measured in MPRA. However, the interpretation of these findings is minimal and somewhat indirect. The manuscript would benefit from further clarification of whether these positions are observed in other reporters. Moreover, it would be helpful to include data on the abundance changes of reporters with single mutations at these sites to demonstrate the versatility of PAL-AI.

Response:

C and G nucleotides within the 3' UTR are generally associated with diminished polyadenylation when compared to A and U nucleotides, as seen in Figure 2E&H in Xiang et al. 2024 (PMID: 38460509). We re-analyzed the data for reporter libraries N60-PAS^{mos} and CPE^{mos}-N60 in the same study and confirmed that for most positions, C was the least favorable nucleotide for polyadenylation (Fig. R1). In addition, at some positions, mutating a C nucleotide to one of the alternative nucleotides either improved the flanking nucleotide contexts of a PAS or a CPE (for example, position -35 in *tpx2.L*) or generated new CPEs (for example, position -72, -36, and -28 in *mad2l1.L*), thereby increasing polyadenylation.

Fig. R1: Relative tail-length difference associated with each nucleotide at each position in the frog oocyte-injected reporter library N60-PAS^{mos} (left) and CPE^{mos}-N60 (right), comparing between 0 and 7 h post-progesterone treatment.

With regard to the abundance changes, both endogenous mRNAs and injected mRNAs are stable in frog oocytes and early embryos until mid-blastula transition (PMID: 7196332, PMID: 9819439, PMID: 38460509, PMID: 34213414). Tail-length changes during this developmental period primarily affect mRNA translation without causing mRNA degradation. Therefore, PAL-AI's power in predicting tail-length changes enables it to predict changes in translational efficiency but not mRNA abundance. We have clarified this in the revised manuscript.

2. The dataset used for training the linear regression model, and PAL-AI is insufficiently described in the figures and text. Readers would find the study easier to follow if Figure 1 included a schematic diagram illustrating the experimental design and the training process of the machine-learning models.

Response:

As requested, we added figures depicting the experimental procedure (Fig. 1a), an overview of the training and testing workflow (Fig. 1b), and a comparison between the linear regression model and the neural network model (Fig. 1d). We also added more description in the corresponding figure legends.

3. Figure 6b shows no compelling reason to restrict the analysis to variants from the All of Us exome or gnomAD databases. Expanding the analysis to all transcript positions expressed in the human germline would reduce unnecessary complexity and improve the interpretability of the results.

Response:

Inspired by the referee's comment, we examined all positions within the last 100 nucleotides of the 3' UTR of mRNA isoforms expressed in human oocytes and binned them by their average predicted impact on tail-length changes (new Fig. 7a, new Supplementary Fig. 9a). Nucleotides predicted to cause the most disruptive tail-length reductions had significantly higher phyloP scores than those with moderate predicted effects, which in turn had significantly higher scores than those with mild effects. For predicted gain-of-function mutations that increase tail lengths, we observed a similar trend, albeit with a smaller effect size.

4. In Figure 6c, the observed effect size seems too small, given the potential confounding factors related to the innate sequence bias of PAL-AI predictions. A more effective approach might involve comparing the ratio of singletons to more prevalent variants across several intervals of PAL-AI-predicted tail length changes. This would provide a clearer and more direct representation of the effect size.

Response:

We performed the requested new analysis. We calculated the fraction of alleles in each bin of different tail-length changes for singletons and frequent alleles with different frequency cutoffs. We then calculated the ratio of the fractions (relative risk) between the singletons and frequent alleles in each bin (normalized to 1 in each bin) and performed Fisher's Exact Tests. Analyses of both the All of Us Research Program and the gnomAD datasets suggested that frequent alleles are less likely to be associated with disruption of tail lengthening (reduced tail-length change) when compared to singletons across all bins and for different allele-frequency cutoffs (new Fig. 7c, d).

Reviewer #2 (Remarks to the Author):

In this work, Xiang and Bartel presented a deep learning model named PAL-AI to predict poly(A) tail length changes in maturing oocytes. They showed that the PAS and CPE elements are major determinants of poly(A) tail lengthening. They performed the reporter assay by injecting mutated mRNAs into oocytes and showed that PAL-AI prediction is correlated with measured changes in mRNAs. The work is novel and valuable as there is no other deep-learning model developed to model cytosolic polyadenylation. I have some comments to improve the work.

1. For the model training, the authors used 1053 nt of 3'UTR sequences. The number looks arbitrary. The authors may try some other longer lengths such as 1200 nt, 1500 nt, and 2000 nt to show the robustness of algorithm performance.

And for the deep learning model parameters in Fig. S1e, why do the authors use these specific filter sizes, kernel sizes, GRU units, and Dense units? The authors may try a few other numbers to show their selected parameters are optimal.

Response:

The 3'-UTR length of 1053 nt was initially chosen due to constraints in early iterations of model architectures. The current PAL-AI model does not have such limitations. We have since tested the model's performance on inputs with various lengths of the 3' UTR, ranging from 100 nt to 5000 nt (new Fig. 1f, g). The model performed robustly, with Pearson R above 0.6 when the last 100 nt sequence of the 3' UTR was included in the input, and with Pearson R above 0.7 when the last 300 nt sequence of the 3' UTR was included. The model's performance increased as longer 3'-UTR sequences were included, until 2000 nt, at which point no further significant improvement was achieved by PAL-AI (new Fig. 1f, g). We have thus chosen 2000 nt as the input sequence length.

To further optimize PAL-AI, we performed an exhaustive search of hyperparameters, including the filter number and the kernel size for the convolution layer, the repeat number of the convolution module, the choice and the unit number of the recurrent module, the unit number of the dense layer, the choice of the activation function, the choice of the loss function, the dropout rate, the L1 and L2 regularization values, the learning rate, the batch size, and the epochs for training. We have added figures evaluating the model's performance when varying these parameters (new Supplementary Fig. 2a, b) to justify the selection of hyperparameter values.

2. The authors showed that the canonical PAS AAUAAA promotes poly(A) tail lengthening. What about other PAS variants, such as AUUAAA, AGUAAA, and others? The authors can mutate AAUAAA to other variants in silico to examine the changes in the predicted values and use the reporter assay data to provide quantitative numbers comparing different PAS signals.

Response:

In Xiang et al., 2024 (PMID: 38460509), it was shown in Figure S2I that both AAUAAA and AUUAAA can promote cytoplasmic polyadenylation in the CPE^{mos}-N60 library. Additional 6-mers shown in that figure were also associated with tail lengthening, but they appeared to be partial AAUAAA or AUUAAA motifs. To avoid tail-lengthening signals from spilling over from the top two 6-mers (AAUAAA and AUUAAA), we excluded all variants that contained either an AAUAAA or an AUUAAA and recalculated the difference in mean poly(A)-tail length in the same library. None of the remaining 6-

mers (which included all of the non-canonical PAS variants) were associated with tail lengthening (new Supplementary Fig. 3h).

To examine the PAL-AI-predicted impact of non-canonical PAS variants, we performed *in silico* mutagenesis and motif-insertion analysis. None of the non-canonical PAS variants, when inserted within 3' UTRs, substantially impacted cytoplasmic polyadenylation, as predicted by PAL-AI (Supplementary Fig. 3f). In addition, other than AUUAAA, the single-nucleotide substitutions of AUUAAA found to be least disruptive of tail lengthening in the cytoplasm did not correspond to the noncanonical PAS motifs previously identified in the context of cleavage and polyadenylation in the nucleus (Supplementary Fig. 3g).

In addition, as suggested, we compared model-predicted differences in tail-length change to experimentally measured values for non-canonical PAS variants in the single-nucleotide mutagenesis library. They showed good correspondence (Fig. 4c).

Together, we found that the previously annotated non-canonical PAS variants promoted little if any cytoplasmic polyadenylation during oocyte maturation.

3. For the reporter assays, the authors showed that the predicted values are systematically shorter than measured tail-length changes. The authors discussed possible reasons. For the reporters, the starting RNAs all contain 35 As, which is different from endogenous genes with different lengths of starting As. Maybe the authors can add this possibility.

Response:

Although frog endogenous mRNAs exhibit varying initial poly(A) tail lengths, their distribution is relatively tight, with a median value similar to that of the injected reporter mRNAs. In Xiang et al., 2024 (PMID: 38460509), the tail length of the reporter mRNAs was selected to match this median endogenous value. Moreover, when we incorporated the starting tail length as an additional input in our model training for frog endogenous mRNAs, we observed no improvement in predictive performance. Thus, we conclude that the initial tail length of the reporter mRNAs is unlikely to be a major cause of the systematic discrepancy between predicted and measured tail-length changes.

In contrast, including starting tail lengths as an input did improve model performance for human endogenous mRNAs. This enhancement was driven by a subset of mRNAs with long poly(A) tails in the GV stage, which undergo subsequent deadenylation in the MII stage. However, since the starting tail lengths of many endogenous mRNAs remain unmeasured, using this feature as a model input would significantly restrict its

applicability. For this reason, we opted to rely solely on sequence information as the model input.

4. The authors discussed that some genes show differences comparing the predicted vs. measured values. One possibility is that the training sequences are biased and do not cover enough cases for these discrepancy cases to train a robust model. The authors can pool together the endogenous genes and reporter RNAs, retrain a model, and use the new model to evaluate the performance. This may help the author obtain a more robust model to cover some rare sequence cases explaining cytosolic polyadenylation.

Response:

Inspired by this suggestion, we modified the original PAL-AI model's architecture and changed the last Dense layer from a single output to multiple outputs (new Fig. 3c). This allowed us to pool the endogenous mRNAs and injected N60(LC)-PAS^{mos} library mRNAs for training and testing. We trained a new model on alternating batches of the two datasets. The resulting model improved its predictions on the injected N60(LC)-PAS^{mos} library mRNAs while maintaining its performance on endogenous mRNA (new Fig. 3d, e). Furthermore, the new model demonstrated superior performance compared to the single-output model on eight out of ten mRNAs in the single-nucleotide mutagenesis library (new Fig. 4b), suggesting it learned additional features that were not present in the endogenous mRNAs.

5. For the genetic analysis, the authors showed selection pressure to avoid genetic variants altering the key motifs mediating cytosolic polyadenylation, such as AAUAAA and UUUUA. But these two motifs are also crucial for nuclear polyadenylation. UUUUA can be bound by Fip1 in the nucleus. The authors need to discuss this. The genetic constraints are not necessarily due to cytosolic polyadenylation.

Response:

The PAS does indeed play an essential role in nuclear pre-mRNA cleavage and polyadenylation. To address this, we repeated our analyses of the All of Us Research and the gnomAD datasets after excluding variants that would introduce or ablate this element.

For the CPE, it partially overlaps with Fip1-binding motifs, which are U-rich, but the overlap is not complete. Thus, some of the observed selective pressure may reflect

constraints acting on Fip1's role in nuclear cleavage and polyadenylation. To acknowledge this point, we have added a paragraph to the Discussion section of the revised manuscript, emphasizing this potential confounding factor and the need for future studies to disentangle relative contributions by different processes to the observed selective pressure.

Reviewer #2 (Remarks on code availability):

Well written with detailed instructions to run the codes.

Reviewer #3 (Remarks to the Author):

The manuscript introduces PAL-AI, a neural network designed to predict changes in poly(A) tail length in maturing frogs, mice, and human oocytes. By considering multiple sequences and contextual features influencing poly(A) tail length, this study offers a potentially robust approach to addressing a critical aspect of gene regulation during oogenesis.

The Authors provide new insights into the role of UGU/GUU-containing motifs in poly(A)-tail lengthening, which has previously been linked to the regulation of translation but not to the tail-length control. The analysis of genetic variants in human populations has potentially important implications for identifying genetic causes of female infertility. This work is timely and well-designed and has the potential to significantly advance our understanding of oocyte maturation and its impact on human fertility.

A notable strength of this study is the initial use of a simple linear regression model, while transitioning to a more sophisticated neural network only when justified by performance shortcomings. This deliberate choice underscores that the adoption of a neural network is a thoughtful decision, not driven by hype for machine learning. The Authors' clear rationale for the variable performance of the model enhances its credibility. While PAL-AI performs well for most mRNAs, it underperforms for some, particularly those with structural features affecting CPE or PAS accessibility. The authors acknowledge this limitation and propose a mitigation strategy in future models. The comprehensive description of the architecture ensures transparency. Furthermore, the authors effectively integrate previously acquired data, construct a robust reporter system, and utilize publicly available datasets, underscoring the versatility of their approach. This thorough validation demonstrates the potential for broad applicability of PAL-AI, given its ability to handle data from different organisms.

However, despite our overall positive impression, the work has several weaknesses that should be addressed before acceptance for publication.

We have one major concern:

The supplementary files available to reviewers needed the source data used to create the figures included in the text (e.g., data presented on scatterplots, results of go-term analysis, coordinates of individual motifs) and the complete input datasets passed to the neural network. With this, the reviewer cannot validate the biological data. This is the most serious deficiency that should be corrected if the work is to be published.

Response:

We have now deposited all Python and R scripts associated with our paper in GitHub (https://github.com/coffeebond/PAL-AI_paper), and we provide the source and processed files necessary to reproduce all figures at Gene Expression Omnibus (GSE280422) and Zenodo (10.5281/zenodo.15461000).

We have also some minor comments:

1. The manuscript lacks information on the validation of PAL-AI training, both in the supplementary materials and the main text. Including a confusion matrix, as well as loss and accuracy curves, would provide a more accurate assessment of the model quality and help identify potential overfitting.

Response:

Because PAL-AI is a regression model, a confusion matrix, which is commonly used for classification tasks, is not an appropriate metric for evaluating its performance. Instead, we assessed model accuracy using scatter plots that compare predicted and experimentally measured values across all datasets. Each plot includes both Spearman's and Pearson's correlation coefficients to quantify prediction accuracy.

In the revised manuscript, we have now included additional plots (new Supplementary Fig. 2c, d) that show the training and evaluation process, including the loss (mean squared error) and metric (mean absolute error) values for both the training and validation datasets across all training epochs. To prevent overfitting, we also implemented an early-stopping callback based on the validation loss.

2. Lines 841-845: It needs to be clarified why the Authors, after training and

validating many models, did not choose the most optimal one but decided to average the results from multiple models. The text lacked a convincing explanation for this strategy.

Response:

We used 10-fold cross-validation to ensure robust model evaluation, which inherently results in slight variability in predictive performance across folds. Additionally, stochastic elements in model training, such as random weight initialization, introduced further variation in performance across repeated training-validation cycles, even within the same fold. Although the overall performance remained consistently high, relying on a single model could introduce fold-specific or initialization-specific biases. To address this, we selected the best-performing model from each fold and used an ensemble of these 10 models to generate final predictions. This strategy helps to average out individual model noise and improve overall generalizability. We have clarified and expanded on this rationale in the revised Methods section.

3. To date, numerous datasets examining the dynamics of poly(A) tails in the germline using long-read sequencing have been published. Comparing the results obtained in this paper with available data from PacBio or Oxford Nanopore would provide orthogonal validation for tail length measurements. Specifically, comparing results with direct RNA sequencing data, which are free of amplification bias, would be a particularly valuable validation of the proposed model.

Response:

To provide orthogonal validation of our tail-length measurements, we compared our results with datasets obtained using long-read sequencing technologies. Specifically, we analyzed data from Lee et al. (2024, PMID: 38306272) that measured poly(A)-tail lengths of mouse oocyte mRNAs using Oxford Nanopore Technologies (ONT). The changes in tail length between GV and MII oocytes, as determined by our PAL-seq method, showed strong correlation with those measured by ONT (new Supplementary Fig. 8e). Notably, PAL-AI predictions correlated even more strongly with the ONT-derived data than they did with PAL-seq measurements (new Fig. 6a, Supplementary Fig. 8d), although the predicted tail-length changes were of smaller magnitude, which is at least partially attributable to systematic differences between the ONT data and the PAL-seq measurements that were used to training PAL-AI (new Supplementary Fig. 8e).

Additionally, poly(A)-tail lengths in human oocytes were measured using PacBio sequencing in a separate study (Liu et al., 2023, PMID: 36646905) and these

measurements also correlated well with PAL-AI predictions (revised Fig. 6b). These analyses provide strong orthogonal support for the accuracy and generalizability of our model.

4. This study analyses data from evolutionarily distant species (frog, mouse, human) using a neural network model trained exclusively on frog data. If the goal was to create a universal model, the training set design should have been more carefully considered. Restricting training to frog data introduces errors in mammalian-specific nucleotide contexts. Alternatively, organism-specific models or at least amphibian and mammalian variants should be developed. Scatterplots indicate the amphibian-trained model underperforms in mammalian cells (Fig. 1a vs. 5a, b). Although PAL-AI shows slightly higher correlations than the initial regression-based approach, this may stem from overplotting (tight data grouping) rather than a significant trend. Using species-specific or taxon-specific models would enable a more detailed investigation of potential patterns in the analyzed data.

Response:

To address this concern, we revised the architecture of PAL-AI by modifying its final layer to support species-specific predictions. We then retrained the model using a pooled dataset comprising frog, mouse, and human sequences, with alternating batches from each species to maintain balance. This multi-species model was subsequently fine-tuned through hyperparameter optimization.

The updated model showed marked improvements in predictive performance for both mouse and human datasets, while maintaining similar accuracy on frog data. With these enhancements, our model better captures species-specific regulatory features.

5. Was the data used to train the model curated? Specifically, were estimates of the relative positions of potential motifs compared to data available in databases/repositories? How reliable are the 3'UTR sequences that were used as references?

Response:

Yes, the data used to train PAL-AI were curated to ensure accurate 3'-UTR annotations. For frog and mouse oocytes, we used PAL-seq data to annotate unique mRNA 3' end isoforms, while for human oocytes, we used publicly available PacBio long-read

sequencing data. The full annotation pipeline and supporting methods are described in Xiang et al., 2024 (PMID: 38460509).

To assess the reliability of our human poly(A) site annotations, we cross-referenced them with PolyA_DB v3 (PMID: 29069441). Of the 17,778 high-confidence poly(A) sites we annotated, 90.8% (16,138) were also present in PolyA_DB. Moreover, after applying an expression cutoff for tail-length analysis, 96.6% (1,569 out of 1,625) of the poly(A) sites used in model training were supported by PolyA_DB annotations. The remaining sites may represent oocyte-specific polyadenylation events not captured in databases that often rely on bulk tissue measurements.

To further validate our annotations, we examined the positional density of known regulatory motifs near poly(A) sites. As expected, we observed a peak of the canonical PAS motif (AWUAAA) approximately 21 nt upstream of the cleavage site (i.e., 15 nt from the 3' end), consistent across human, mouse, and frog, and in agreement with prior studies (PMID: 15647503; 22454233; 28525757) (Fig. R2).

These findings collectively support the reliability of the reference sequences used to train PAL-AI.

Fig. R2: Positional densities of the PAS and the CPE along the last 500 nt of human, mouse, and frog mRNA 3' UTRs. The insets show the positional densities in the last 50 nt of the 3' UTR.

Reviewer #3 (Remarks on code availability):

1. The software is available at the provided link and can be installed without issues. It includes a test dataset that enables users to verify its functionality. However, when analyzing the test data, the INN_main.py script generates a warning at line 733. While this warning does not directly affect the predictions, it suggests that the check for 'NA' values may not be functioning as intended. Addressing this issue will ensure the script handles 'NA' values in the Y_df['y'] column correctly.

Response:

In the new version of the code, we have fixed this issue.

2. Adding a repository section explaining each column in the final output (classification results) or using more intuitive table headers would help audiences less familiar with machine learning outputs. This would also simplify downstream processing for general users.

Response:

We have added additional information regarding the output files and expanded the instructions on how to use our code.

3. The use of the SELU function (instead of the commonly used RELU) is clever, allowing it to counteract vanishing and exploding gradient problems and, thus, a decrease in the quality of model predictions. However, has weight initialization been implemented properly? The SELU function requires a specific kernel initializer, as pointed out in the TensorFlow docs. Has the dropout type also been adjusted? The code available in the GitHub repository uses the default initializer and dropout, which is not a very fortunate choice if one does not want SELU to lose its self-normalizing properties.

Response:

In our initial implementation, we observed empirically that the SELU activation function paired with the He normal initializer yielded better performance than when paired with the recommended LeCun normal initializer, though we did not fully understand why. At that time, we also did not test alternative dropout layers, such as AlphaDropout, which are designed to preserve the self-normalizing properties of SELU.

In response to this comment, we performed a more systematic comparison of activation functions, kernel initializers, and dropout types. The three configurations tested are summarized below.

Option	Activation function	Kernel initializer	Dropout layer
1	SiLU	He normal	Dropout
2	SELU	LeCun normal	Alpha Dropout
3	Leaky ReLU	He normal	Dropout

We evaluated these combinations in the context of broader hyperparameter tuning and found that Option 3—Leaky ReLU with He normal initialization and standard Dropout—consistently yielded better performance. As a result, we adopted this architecture in the revised PAL-AI model. These updates are reflected in the revised Methods section, and the updated code has been committed to our GitHub repository.

4. The pipeline supports an alternative network architecture, ResNet, but this is not mentioned in the manuscript. The absence of a comparison between the described architecture and ResNet results is a notable omission and leaves the reviewer somewhat dissatisfied.

Response:

During the early development of PAL-AI, we developed a version that incorporated ResNet in place of the convolutional module. While this architecture occasionally achieved comparable predictive performance, its results were less consistent and it required substantially more parameters (on the order of millions, 10-fold more) than what was used in the current PAL-AI model. In addition, training the ResNet-integrated model would require significantly more computational resources, making it less desirable for iterative development. Given these limitations, we decided not to pursue this approach further. We have now added this design consideration in the text and included a figure comparing the performance between PAL-AI and the ResNet-based model (new Supplementary Fig. 2e).

Reviewer #4 (Remarks to the Author):
